# Evidence from oyster suggests an ancient role for Pdx in regulating insulin gene expression in animals

Fei Xu [1,2 ✉], Ferdinand Marlétaz [1,3,4], Daria Gavriouchkina [3,5], Xiao Liu[6], Tatjana Sauka-Spengler [5], Guofan Zhang[2,7] & Peter W. H. Holland [1]

Hox and ParaHox genes encode transcription factors with similar expression patterns in divergent animals. The *Pdx* (*Xlox*) homeobox gene, for example, is expressed in a sharp spatial domain in the endodermal cell layer of the gut in chordates, echinoderms, annelids and molluscs. The significance of comparable gene expression patterns is unclear because it is not known if downstream transcriptional targets are also conserved. Here, we report evidence indicating that a classic transcriptional target of Pdx1 in vertebrates, the *insulin* gene, is a likely direct target of Pdx in Pacific oyster adults. We show that one *insulin*-related gene, *cgILP*, is co-expressed with *cgPdx* in oyster digestive tissue. Transcriptomic comparison suggests that this tissue plays a similar role to the vertebrate pancreas. Using ATAC-seq and ChIP, we identify an upstream regulatory element of the *cgILP* gene which shows binding interaction with cgPdx protein in oyster hepatopancreas and demonstrate, using a cell culture assay, that the oyster Pdx can act as a transcriptional activator through this site, possibly in synergy with NeuroD. These data argue that a classic homeodomain-target gene interaction dates back to the origin of Bilateria.

[1] Department of Zoology, University of Oxford, Oxford, UK. [2] Key Laboratory of Experimental Marine Biology, Center for Ocean Mega-Science, Institute of Oceanology, Chinese Academy of Sciences, Qingdao, China. [3] Molecular Genetics Unit, Okinawa Institute of Science and Technology Graduate University, Okinawa, Japan. [4] Centre for Life's Origins and Evolution, Department of Genetics, Evolution and Environment, University College London, London, UK. [5] Radcliffe Department of Medicine, Weatherall Institute of Molecular Medicine, University of Oxford, Oxford, UK. [6] Fishery College, Key Laboratory of Marine Fishery Equipment and Technology of Zhejiang, Zhejiang Ocean University, Zhoushan, China. [7] Laboratory for Marine Biology and Biotechnology, Qingdao National Laboratory for Marine Science and Technology, Qingdao, China. ✉email: xufei@qdio.ac.cn

The ParaHox genes comprise the Gsx, Pdx and Cdx homeobox gene families, each of which can be traced to the most recent common ancestor of the extant bilaterally symmetrical animals, the Bilateria[1,2]. Like their better known paralogues, the Hox genes, ParaHox genes were ancestrally arranged in a gene cluster and a tightly linked genomic arrangement is still retained in amphioxus (*Branchiostoma floridae*)[1], sea star (*Patiria miniata*)[3], acorn worm (*Ptychodera flava*)[4], two molluscs[5,6] and many vertebrates. Genome duplications in vertebrates generated additional ParaHox genes, including a second *Pdx* gene in coelacanth and sharks[7]. Also reminiscent of Hox genes, the central ParaHox cluster gene *Pdx* is primarily expressed in a spatial pattern that marks a zone of the body, in this case a region of the developing endoderm in vertebrates, amphioxus, echinoderms, acorn worm, annelid worm and a gastropod mollusc[1,3,4,8–14]. This expression is thought to specify the fate of a central endoderm zone and, consistent with this view, homozygous inactivation or deletion of *Pdx1* in mice or humans results in absence of the pancreas and anterior duodenum[9,10,15–17]. In vertebrates, spatial expression persists into the adult where *Pdx1* is expressed in beta-cells of the pancreatic islets and is a direct transcriptional activator of several targets including the *insulin* gene[18].

Although Hox and ParaHox gene expression is broadly similar between phyla, it is not clear whether their transcriptional interactions are conserved. The upstream processes that establish Hox gene expression boundaries are different between vertebrates and insects, and not well understood for ParaHox genes. Downstream of Hox and ParaHox genes there is also little indication of deeply conserved targets thus far. This contrasts with a homeobox gene of the PRD class, *Pax6*, for which some ancient conserved targets are known, including *Mab21* (in nematode and mouse), an *insulin-like peptide* gene (in *Drosophila* and mouse)[19] and *crystallin* genes (in scallop and mouse)[20]. To gain insight into the ancestry of vertebrate genetic regulatory processes, ideally, we must make comparisons to the other great clades of bilaterian animals, the Ecdysozoa and the Lophotrochozoa. Characters that are homologous between vertebrates and either of these clades must date back to the base of the Bilateria. It is notable, however, that members of the Lophotrochozoa share more genome characters in common with Deuterostomia (including vertebrates), because of extensive divergence and secondary loss of genes in Ecdysozoa[21]. The *Pdx* gene is a good example, having been secondarily lost in almost all Ecdysozoa, including *Drosophila* and *Caenorhabditis*[2]. We therefore investigated whether the Pdx might be a direct regulator of an *insulin* gene in a member of the Lophotrochozoa, the Pacific oyster *Crassostrea gigas*. If the oyster *Pdx* does regulate *insulin* gene expression then it is likely this target gene interaction dates to the base of Bilateria. A practical difficulty with testing this hypothesis is that the evolutionary history of *insulin*-related genes is complex, with numerous duplications and losses across animal evolution. *Insulin*-related genes in the genome of Pacific oyster have been well characterised and one gene, *cgILP*, is suggested to be a lophotrochozoan orthologue of *insulin/IGF*[22]. In this study, we further analyse the expression of *cgILP* using quantitative RT-PCR and in situ hybridisation, and use an Assay for Transposase-Accessible Chromatin with high throughput sequencing (ATAC-seq) to uncover putative regulatory elements containing predicted Pdx-binding sites. Chromatin immunoprecipitation suggests binding of Pdx protein, and luciferase assays in cell culture verifies the ability to activate gene expression through this non-coding site 1.8 kb upstream of the oyster *cgILP* gene. These data argue that regulation of insulin gene expression by Pdx dates back to the origin of Bilateria.

## Results

**Co-expression of *cgILP* and *cgPdx* in oyster digestive tissue**. We conducted further phylogenetic analyses on the insulin-like peptides identified previously in the Pacific oyster genome[22–24]: *cgMIP123*, *cgMIP4*, *cgMILP7* and *cgILP*, incorporating sequences from across the Metazoa (Supplementary Data 1 and Fig. 1). The result is generally consistent with previous reports[22,25], with minor topological differences in tree inference. The four oyster genes analysed do not group closely together, unlike the multiple insulin-related genes reported from spider (*Stegodyphus*) and silkmoth (*Bombyx*) (Fig. 1; Supplementary Fig. 2). Two of the oyster genes (*cgMIP123* and *cgMIP4*) consistently group with the Molluscan Insulin Peptide (MIP) group and are physically linked indicative of tandem duplication, as reported previously[22]. *cgMILP7* is a likely orthologue of *Drosophila*[22], beetle and termite *ILP7*, a conclusion also supported by their predicted chain structure (Supplementary Table 1). Precise tree topology is dependent on alignment and sampling, and although *cgILP* is placed less precisely in our analysis, we find that it groups with chordate insulin/IGF genes, Bombyxins and spider genes (Fig. 1), supporting the close relationship with insulin[22].

The simplest explanation for these results is that there was an early diversification of insulin-related genes in metazoan evolution[22], followed by loss of genes in many taxa including vertebrates. *cgILP* is the putative orthologue of vertebrate insulin/IGF genes.

A previous report revealed digestive gland expression of *cgILP* and proposed the possible involvement in the control of digestion and gametogenesis[22]. We further conducted realtime quantitative reverse transcription PCR (qPCR) and in situ hybridisation and confirmed that only *cgILP* has expression specific to the endodermal cell layer of oyster hepatopancreas and stomach (Fig. 2a–j), comparable to the endodermal expression of vertebrate *insulin*[22]. Analysis on the oyster *cgPdx* homeobox gene further revealed the overlap with *cgILP* in the endodermal lining of digestive tissues (Fig. 2a, k–m). High expression of *cgILP* and *cgPdx* in the digestive gland (mainly composed of hepatopancreas) is consistent with previous studies[22,23] (Supplementary Table 2).

**Sequence characteristics of oyster Pdx protein**. Alignment of Pdx amino acid sequences from vertebrates, amphioxus and bivalves highlights strong conservation of the homeodomain and hexapeptide, and weaker but clear conservation of an N-terminal transcriptional activation domain[26] (Supplementary Fig. 3). Amino acids at several key sites, where mutations are associated with human diabetes (e.g. Q59L and P33T), are conserved in the oyster.

**RNAseq analysis of oyster hepatopancreas**. Expression of oyster *cgILP* and *cgPdx* genes in hepatopancreas parallels co-expression of vertebrate *insulin* and *Pdx1* in the pancreas, but arguments for homology would be stronger if hepatopancreas had functional similarity to vertebrate pancreas or midgut. As previous RNAseq from oyster digestive gland[23] included hepatopancreas and other tissues, we conducted RNAseq specifically on oyster hepatopancreas tissue. We identified 1527 hepatopancreas-enriched genes by comparing to transcriptomes from other oyster tissues (Supplementary Data 2). Gene ontology analysis reveals enrichment in genes encoding enzymes (deaminase, hydrolase, lyase, oxidoreductase, and transferase, plus enzyme activators and inhibitors), compatible with a digestive role (Supplementary Fig. 4). Included were homologues of several digestive enzymes produced by the vertebrate pancreas notably trypsin (NCBI accession:

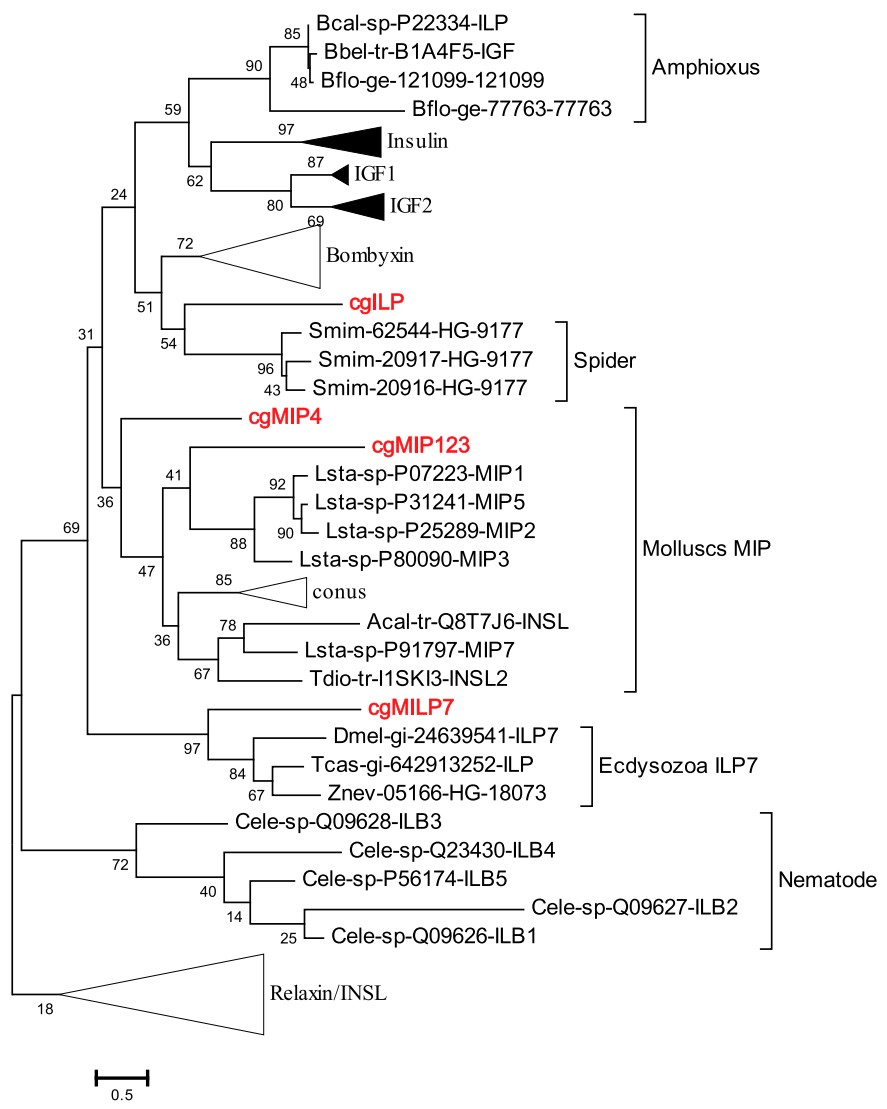

**Fig. 1 Phylogenetic analysis of insulin-related peptides (RAxML method).** Species: Bcal *Branchiostoma californiense* (California amphioxus), Bbel *B. belcheri tsingtauense* (Chinese amphioxus), Bflo *B. floridae* (Florida amphioxus), Cgig *Crassostrea gigas* (Pacific oyster), Smim *Stegodyphus mimosarum* (spider), Lsta *Lymnaea stagnalis* (snail), Acal *Aplysia californica* (sea hare), Tdio *Tritonia diomedea* (sea slug), Dmel *Drosophila melanogaster* (fruitfly), Tcas *Tribolium castaneum* (beetle), Znev *Zootermopsis nevadensis* (termite), Cele *Caenorhabditis elegans* (nematode). Some species are covered by collapsed sub-trees, see Supplementary Data 1, Figs. 1 and 2 for details.

LOC105348513), chymotrypsin (LOC105331575), pancreatic triacylglycerol lipases (LOC105329552, LOC105329276, LOC105340853, and LOC105344176), pancreatic lipase-related protein (LOC105331423, LOC105331427, and LOC105331424), pancreatic alpha-amylase (LOC105342874), and other alpha-amylase-like genes (LOC105348751 and LOC105348748).

In addition to *cgILP*, several other genes were related to insulin function. These include *Neuroendocrine Convertase 1-like* (*cgPCSK1*, LOC105344671) possibly orthologous to the PC1/3 enzyme that cleaves pro-insulin to liberate C-peptide[27], *X-Box-Binding Protein 1-like* (*cgXBP1*, LOC105344684) thought to mediate insulin action and expressed in human pancreas and liver[28], *Regulatory Factor X6* (*cgRfx6*, LOC105331217) required for differentiation of *insulin*-producing islet cells, and the homeobox genes *NK2 Homeobox 2* (*cgNkx2.2*, LOC105335274) and *Pituitary Homeobox* (*cgPitx*, LOC105337384) orthologous to genes involved in the development of vertebrate pancreas and islet cells[29–31]. *cgPdx* transcripts were present in the hepatopancreas RNAseq dataset but not the organ-enriched subset, due to additional expression in gonad tissues. The latter is inconsistent

with qPCR and may reflect contamination by intestine tissue coiled inside the gonad.

**ATAC-seq profiling of oyster hepatopancreas**. Conservation of the transactivation domain and homeodomain, as well as the overlapping expression of oyster *cgPdx* with *cgILP*, are compatible with direct regulation, comparable to the transcriptional regulation of vertebrate *insulin* by Pdx1. To examine this hypothesis, we first examined the chromatin accessibility landscape in the oyster hepatopancreas using ATAC-seq (Figs. 3 and 4)[32], after optimising methods for dissociating and isolating hepatopancreas cells. After data processing, we obtained 168,206 peaks corresponding to accessible chromatin regions observed reproducibly in two replicates ($r^2 > 0.88$; Supplementary Fig. 5a). Of the called peaks, ~49% were situated in intergenic and intronic non-coding regions, 8.8% within putative promoters and 37.8% within coding exons (Supplementary Fig. 5b, c).

We scanned these accessible chromatin peaks for consensus Pdx motifs obtained through Transcription Factor Binding

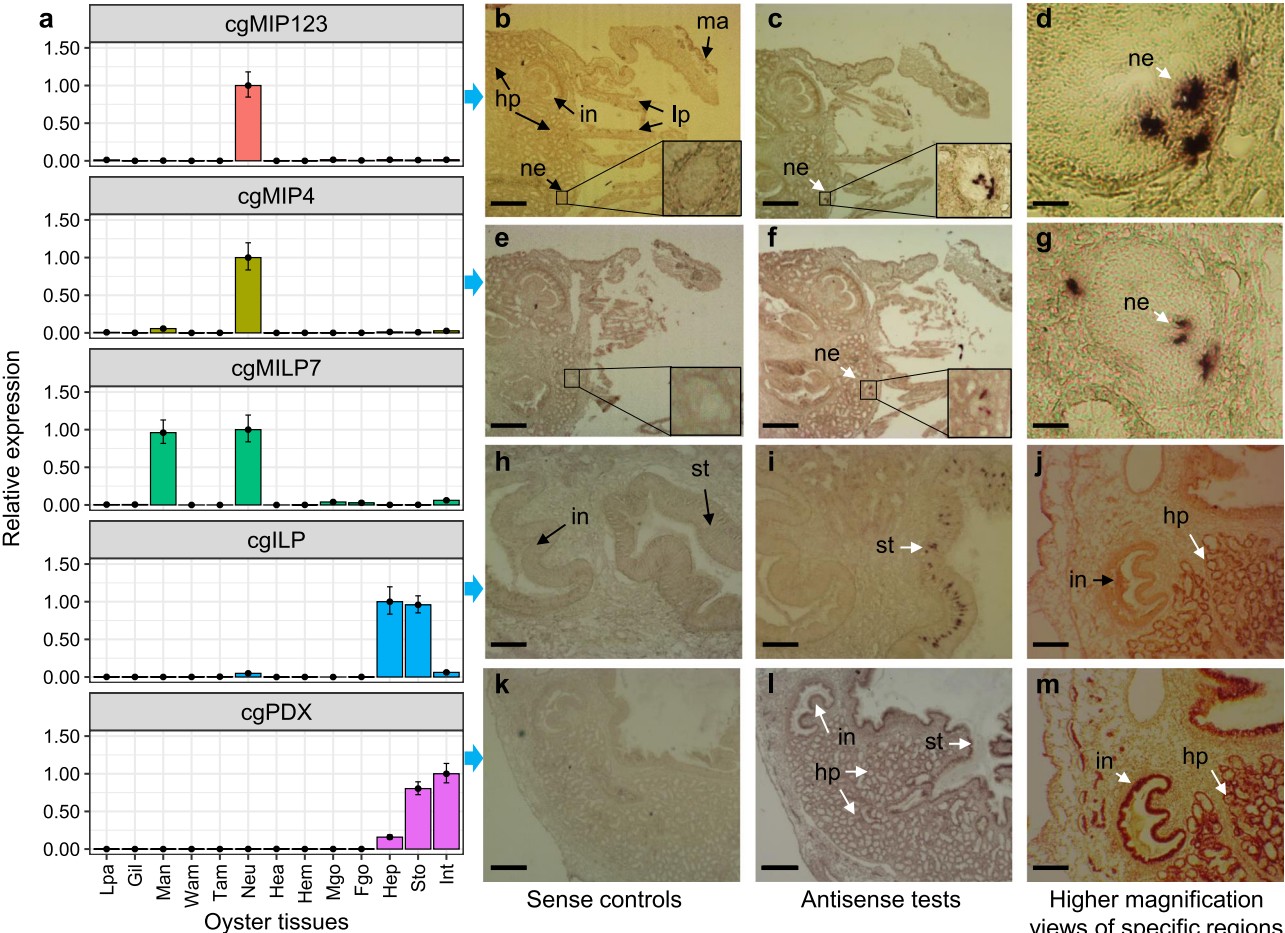

**Fig. 2 Analysis of oyster ILPs and Pdx gene expression. a** qPCR survey of gene expression levels in oyster tissues for *cgMIP123*, *cgMIP4*, *cgMILP7*, *cgILP* and *cgPdx*. Within each gene, qPCR results were normalised to the average value of the tissue with the highest relative quantification (RQ). Error bars represent 95% confidence intervals around the mean of RQs ($n = 1$ cDNA sample mixed from three animals). Three (*cgILP* and *cgPdx*) or four (*cgMIP123*, *cgMIP4* and *cgMILP7*) technical replications were conducted. Lpa labial palps, Gil gill, Man mantle, Wam white adductor muscle, Tam transparent adductor muscle, Neu neuron, Hea heart, Hem haemolymph, Mgo male gonad, Fgo female gonad, Hep hepatopancreas, Sto stomach, Int intestine. Broad blue arrows leading from qPCR panels to rows of images indicate in situ hybridisation analysis using the indicated genes (**b**–**m**). White arrows point to sites of gene expression. Adjacent sections were used to compare sense controls (**b**, **e**, **h**, **k**) with antisense tests (**c**, **d**, **f**, **g**, **i**, **j**, **l**, **m**), to compare expression between *cgMIP123* and *cgMIP4*, and to compare *cgILP* and *cgPdx* gene expression. **c**, **d** Adult section showing *cgMIP123* expression in a branchial neuron (possibly the cerebro-visceral connective). **f**, **g** Adjacent sections showing *cgMIP4* expression in identical sites to *cgMIP123*. **i**, **j** Sections showing *cgILP* expression in the stomach wall and ducts of the hepatopancreas. **l**, **m** Adjacent sections showing *cgPdx* expression in identical sites to *cgILP*, except for additional staining of *cgPdx* RNA in parts of the gut wall (compare panel **m** to **j**). Abbreviations (pointed by black arrows): st stomach, in intestine, lp labial palps, ma mantle, ne neuron, hp hepatopancreas. Bar under each panel marks, 500 μm (**b**, **c**, **e**, **f**, **h**, **i**, **k**, **l**), 250 μm (**j**, **m**), 15 μm (**d**, **g**). Representative of at least two independent experiments for **b**–**m**. Source data are provided as a Source Data file.

Site (TFBS) clustering of five Pdx1 TFBSs derived from the literature (Supplementary Fig. 6). A total of 19,104 putative Pdx-binding sites were deemed statistically significant within the 168,206 accessible chromatin peaks ($p < 0.05$). The number is comparable with observed binding peaks in human pancreatic islet cells analysed with ChIPseq (18,294)[33] or ATAC-seq (~6000)[34]. A total of 795 oyster hepatopancreas-enriched genes were detected to have putative Pdx-binding sites in associated ATAC-seq peaks; of these, 56 are orthologous to genes predicted to be Pdx1 targets in previous high throughput studies on human or mouse[35–39] (Supplementary Data 3 and Fig. 7). To further assess the accuracy of binding site enrichment analysis in our dataset, we also employed a set of 623 TFBSs to search all open chromatin peaks, and found that 583/623 were present in our dataset in more than one copy (Fig. 3). TFBSs were uncovered predominantly in promoters rather than enhancers. We did not

detect any other consensus or individual homeodomain motifs overlapping with the predicted Pdx sites, although different homeodomain sites were enriched, confirming the specificity of Pdx-binding site searches (Fig. 3).

Interestingly, we found a broad ATAC-seq peak upstream of *cgILP* gene containing a predicted Pdx-binding site (A-box, TTCTAATTAC at −1806 bp), consistent with a putative *cis*-regulatory role of these regions in *cgILP* regulation. Possible binding sites of other transcription factors were also identified, including a putative E-box (at −1760 bp: CAGTTG) which could potentially bind the bHLH transcription factor NeuroD implicated in insulin gene regulation[40], and a possible hormone response element (HRE at −1735: AGGTCA) which might bind the nuclear receptor HNF4a[41] (Fig. 4). Putative Pdx-binding sites were also identified at ATAC-seq peaks upstream of *cgPCSK1* and *cgXBP1* at positions −85 bp (TGCTAATTGG) and −2448 bp

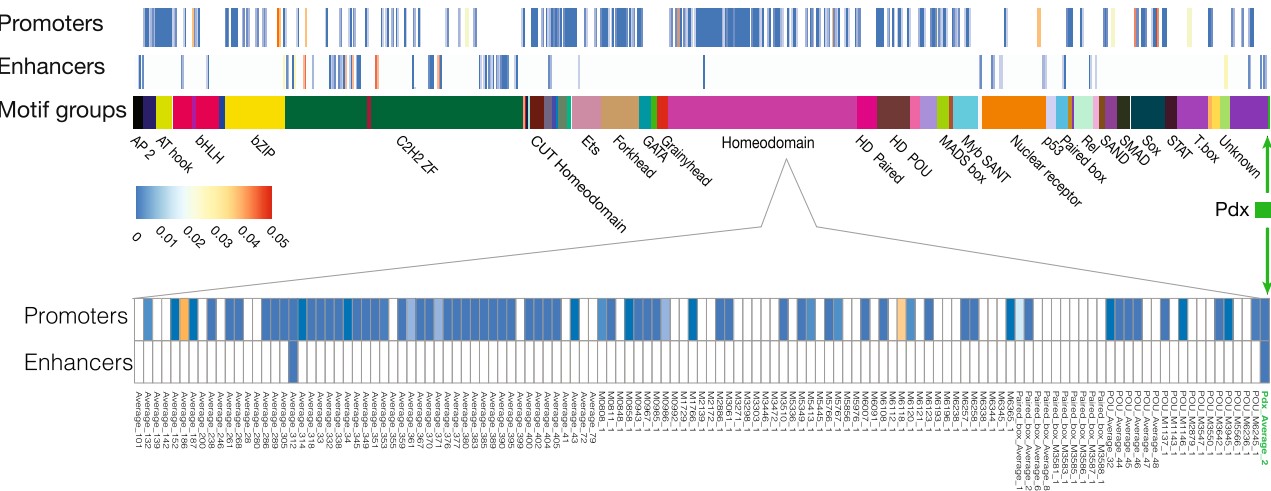

**Fig. 3 Homeodomain TFBS and Pdx consensus motif are enriched in open chromatin promoter and enhancer peaks.** TFBS enrichment test *p* values after Bonferroni correction were assessed for Pdx consensus motif and 583 other consensus motifs.

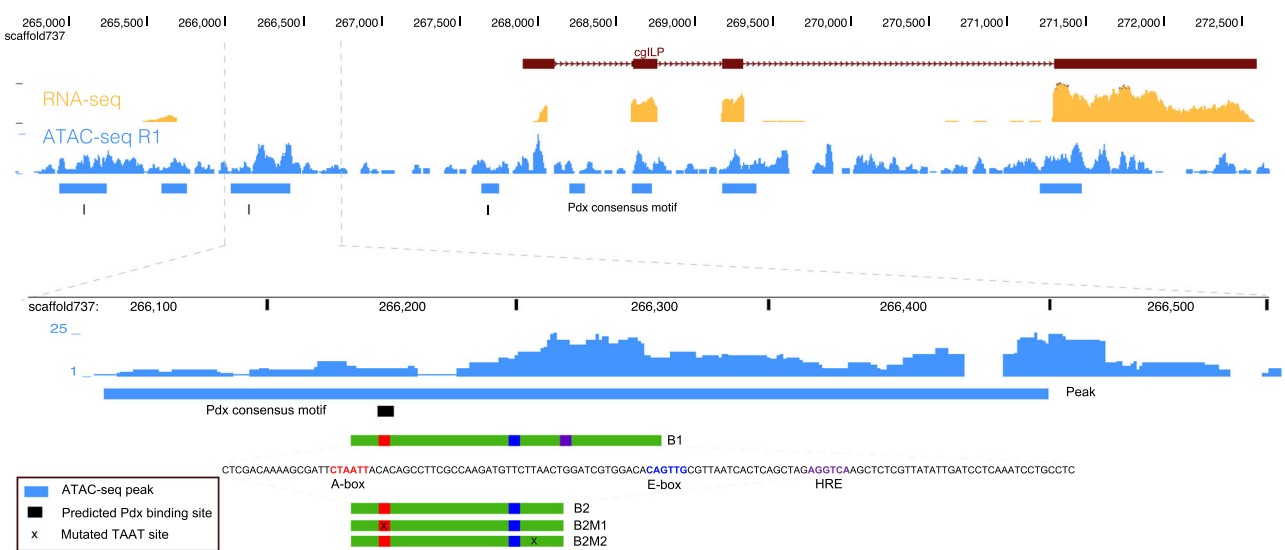

**Fig. 4 Potential transcription factors binding sites (A-box, E-box, and HRE) identified by ATAC-seq peak.** A 72 bp region (green box) was used for dual-luciferase assays. Gene structure of *cgILP* is shown in the top bars, where purple bars indicate exons, and arrows along the line indicate transcriptional orientation. The ATAC-seq peaks are not direct extrapolation of the regions with the highest sequencing signal as the calling process takes into account both replicates, the averaged background over a window, and the shift induced by the library construction process.

(GTCTAATTGA) respectively; these loci are orthologous to genes implicated in pro-insulin processing in mammals (Supplementary Fig. 8).

**Pdx regulation of transcription through a non-coding element.** As oyster cultured cell lines and transgenic approaches are currently unavailable, we conducted dual-luciferase assays in HeLa and COS7 cells to examine if the putative Pdx TFBS upstream of *cgILP* is responsive to oyster Pdx protein. First, we used a vector delivering a fluorescent protein (GFP)-tagged protein to confirm that oyster Pdx is successfully transported to the cell nucleus (Supplementary Fig. 9). Second, we ectopically expressed untagged oyster Pdx protein together with a reporter gene comprising the putative response element (A-box, B1 in Fig. 4) driving expression of firefly luciferase and a transfection control plasmid. Gradient experiments indicated that the activation effect through element B1 was influenced by the quantities of transfected *cgPdx* expression plasmids (Fig. 5a). The activation effect is weak in

HeLa cells, where only 1.2–1.4 fold increase was observed, even when 500 ng Pdx expression plasmid was used (Supplementary Fig. 10a); however, in COS7 cells the transactivation responses are much more robust for both cgPdx and mPdx1 (Fig. 5a and Supplementary Fig. 10b). When the central TAAT was mutated into TACT (B2M1, Fig. 4), the transcriptional activation was decreased. In contrast, mutating a second TAAT site within the region (B2M2) had no effect on the transcriptional activation (Supplementary Fig. 10d, e). These results suggest that the cgPdx protein is capable of transactivating expression through the predicted A-sequence found upstream of the *cgILP* gene.

To test if Pdx activity is enhanced by a co-factor, we also introduced the oyster orthologue of NeuroD, a bHLH protein. A similar dose-dependent, but weaker activity, was observed when a cgNeuroD expression plasmid was transfected into COS7 cells along with the cgILP B1 site reporter gene (Fig. 5b). Co-transfecting cgPdx and cgNeuroD expression plasmids, plus the reporter gene, revealed a synergistic response. The transcriptional activity level of the target reporter gene was significantly higher

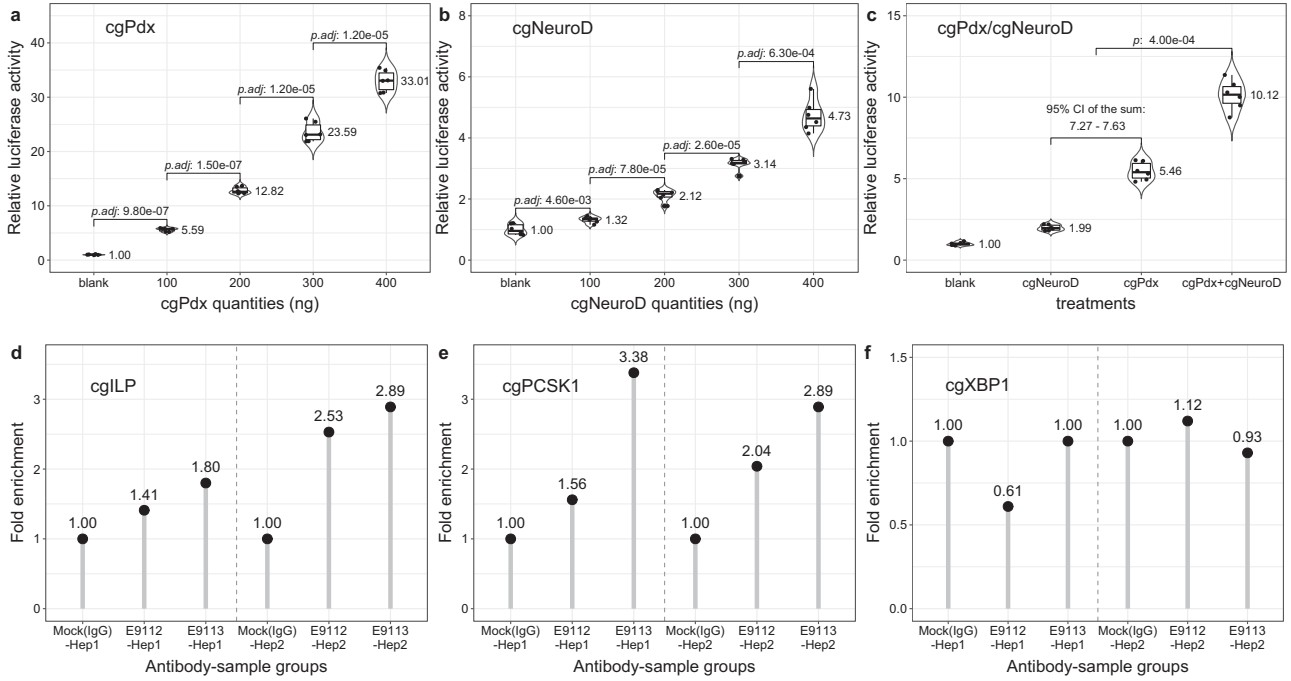

**Fig. 5 Transcriptional activity of oyster *cgPdx* and *cgNeuroD* in cell culture and oyster hepatopancreas tissue. a** Transcriptional activity of oyster *cgPdx* on *cgILP* B1 element in COS7 cells demonstrating dose-dependent induction of luciferase reporter gene expression. Points show values of each individual assay. Violin and box plots are used to show the distribution of the data, with the mean at the right side. *p* values of comparisons (two-sided *t*-test) between adjacent groups are shown. **b** Transcriptional activity of *cgNeuroD* on *cgILP* B1 element in COS7 cells showing a similar but weaker dose-dependent induction effect. **c** Co-transfection of *cgPdx* and *cgNeuroD* gives higher transcriptional activity than the sum of the individual activities, suggesting functional synergy between cgPdx and cgNeuroD proteins. 150 ng of cgPdx and cgNeuroD plasmids was used for group (cgPdx + cgNeuroD). The pool of all the possible sums of individual relative luciferase values between cgNeuroD and cgPdx groups was used to estimate the 95% confidence interval (CI) of the activity sum. Significance of difference between group (cgPdx + cgNeuroD) and the sum of cgNeuroD and cgPdx groups was calculated by bootstrapping ($n = 100,000$). For experiments in **a**–**c**, $n = 3$ biologically independent cells by two technically independent measurements. Box represents a range from the first to the third quartile while whisker shows the minimum and maximum values. The line in each box shows the median. Relative luciferase activity in the cells transfected with blank pSI and the reporter plasmid is set at 1.00. *p* values were calculated by two tailed Student's *t* test and Holm's method to correct for multiple comparisons (*p.adj*). **d**–**f** ChIP-qPCR assays on predicted cgPdx binding sites upstream of three putative Pdx target genes in oyster (*cgILP*, *cgPCSK1*, and *cgXBP1*). Two anti-cgPdx polyclonal antibodies (E9112 and E9113) were used to assay two oyster hepatopancreas samples (Hep1 and Hep2). Source data are provided as a Source Data file.

than the sum of the individual activities when the two plasmids were transfected separately (Fig. 5c). A similar but stronger synergistic effect was also observed when co-transfecting *cgNeuroD* and *mPdx1* expression plasmids (Supplementary Fig. 10c).

To test whether oyster Pdx protein was bound to this site in vivo, we raised two polyclonal antibodies (E9112 and E9113) against cgPdx-specific peptide (1–160 amino acids) and conducted four ChIP-qPCR assays on oyster hepatopancreas tissue. These experiments revealed that DNA at the predicted site around the ATAC-seq peak (B1) had significantly higher fold enrichment (~2.2 times, *p* < 0.05, Fig. 5d) than the mock (IgG). This adds support to the proposal that Pdx protein regulates *cgILP* gene expression in adult oysters. We also tested whether oyster Pdx protein was binding DNA around other putative target genes; we found the putative Pdx-binding site upstream of *cgPCSK1* also showed higher fold enrichment (~2.47 times, *p* < 0.05, Fig. 5e), but the *cgXBP1* site did not (Fig. 5f).

## Discussion

The discovery of homeobox genes with similar expression patterns in widely divergent phyla triggered a revolution in biology. From the mid-1980s onwards, a series of studies questioned the view that body form was built and maintained by different genes in different phyla and supplemented it with a concept stressing evolutionary

modification of a shared 'tool-kit' of genes. However, finding similar gene expression patterns does not demonstrate equivalent function. For some homeobox genes, notably Hox genes, mutation and gene deletion experiments in vertebrates and arthropods have demonstrated comparable roles in specifying regional identity in the embryo, although the suites of downstream target genes regulated by Hox genes have not yet been elucidated. The ParaHox genes are as ancient as Hox genes, sharing a common ancestry dating back to before the divergence of cnidarians and bilaterians[42]. The central ParaHox gene, *Pdx*, is the best studied due to its involvement in insulin regulation and since mutations in this gene are implicated in monogenic type 2 diabetes in humans.

The *Pdx* homeobox gene has embryonic and adult functions. In the development of vertebrates, amphioxus, annelids, molluscs and echinoderms, the gene is expressed in a zone or domain of the endoderm, which in mouse is fated to become the pancreas plus parts of the duodenum, bile duct and stomach[10]. As development proceeds, the expression of *Pdx* (called *Pdx1* in vertebrates) becomes more localised and restricted to pancreatic islets. Within islets, *Pdx1* is a direct transcriptional activator for *insulin*, *somatostatin* and other genes. It is not known how ancient is the regulation of *insulin* transcription by Pdx.

The Insulin superfamily is well understood in vertebrates, and can be divided into an insulin/insulin growth factor (IGF) group and a relaxin/insulin-like peptide (RLN/INSL) group. All possess cysteine residues forming disulphide bonds within and between

peptide chains. Insulin/IGF proteins function mainly as metabolic regulators and mitogenic growth factors; relaxin/INSL proteins have roles in reproduction and neuroendocrine regulation, signalling through G-protein coupled receptors. Comparison between vertebrate and tunicate genomes reveal that insulin/IGF genes diversified through a tandem duplication event followed by genome duplications in vertebrates and further tandem duplications[43]. The picture becomes more confusing as other invertebrates are examined. The silk moth *Bombyx mori* has a large number of insulin-related *Bombyxin* sequences, closely similar to each other, but other insects have quite different genes such as *Drosophila ILP7*. Spiders have a set of duplicate insulin-related genes that look different again, as does the diversity of *insulin*-related genes in nematodes and a greatly expanded set in molluscs, especially venomous cone snails[44].

The existence of insulin-like peptides in a bivalve mollusc, *Mya arenaria*, was actually reported shortly after the discovery of insulin[45]. Cloning and sequencing of a cDNA for a molluscan insulin-related peptide (MIP) was reported in the 1980s[46]. However, this initial *ILP* gene from a snail, and the first reported oyster *ILP* gene[47] were expressed in neurons (the latter is now designated *cgMIP123* gene). Genome sequencing revealed that both the limpet *Lottia gigantea* and the Pacific oyster genome have multiple insulin-related genes, including a γ-type ILP sharing a conserved structure with vertebrate insulin/IGFs[22,25]. Furthermore, Pacific oyster *cgILP* is expressed in the digestive gland[22]. Using qPCR and in situ hybridisation we further studied the expression of this gene in the complex digestive gland of adult oysters, and found that expression of *cgILP* overlaps with the Pdx homeobox gene in the endodermal cell layer of hepatopancreas and other gut tissues. The enrichment of *cgPdx*, *cgILP*, as well as its processing convertase *cgPCSK1* is comparable to endodermal expression of *insulin* and *Pdx1* in vertebrate developing gut and adult pancreas. We note that expression of the two genes is not precisely congruent in oyster, nor indeed in vertebrates. Indeed, Pdx1 contributes to other biological processes in vertebrates, such as activating other genes (*somatostatin* and *glucokinase*). We do not suggest that the complex tube system of the oyster hepatopancreas is homologous 'as an organ' to the vertebrate pancreas[48]. This is because some taxa (e.g. amphioxus and hagfish) lack a clear pancreas, with insulin-producing cells found in the gut or bile duct mucosa[48]. Thus, we suggest that oyster hepatopancreas evolved in parallel to the vertebrate pancreas.

A caveat to the expression similarities between oyster and vertebrates centres on whether the oyster hepatopancreas shares functional parallels with the vertebrate pancreas. The functions of the molluscan hepatopancreas are incompletely understood, with the organ's compound name suggestive of both liver and pancreas-like functions while there are also data supporting immune-related roles[49]. In mammals, liver and ventral pancreas bud have a close lineage relationship, and may arise from the same precursors in the foregut endoderm[50]; indeed, Pdx1 plays key roles during the pancreas differentiation by repressing hepatic genes[35]. We used RNAseq to test whether oyster hepatopancreas has similarities to the vertebrate pancreas, finding that the oyster hepatopancreas transcriptome is enriched in genes typical of the digestive functions of the vertebrate exocrine pancreas (trypsin, chymotrypsin, lipases, amylase) and contains several genes associated with islet differentiation and insulin processing in the endocrine pancreas. These data are suggestive of comparable functions, but even if precise physiological functions differ between oyster and vertebrates there is little doubt that bilaterian midguts are homologous[51,52]. We conclude that the molluscan hepatopancreas and the vertebrate pancreas evolved from a homologous endodermal region expressing *Pdx* and *insulin* in the latest common ancestor of bilaterians.

In vertebrates, Pdx is a key transcriptional activator of the *insulin* gene, with its activity is modulated by co-factors[53]. We used a cell culture system to test if oyster Pdx protein, with or without a bHLH co-factor, can activate gene expression through a putative binding site identified upstream of the *cgILP* gene. The putative binding site (named B1) contains a canonical Pdx-binding sequence and is located in a broad accessible chromatin region revealed by ATAC-seq on oyster hepatopancreas tissue. In cell culture assays, we find that cgPdx can indeed transactivate gene expression through the B1 site found upstream of *cgILP*. Furthermore, when the bHLH protein cgNeuroD was co-expressed along with cgPdx, a greater transcriptional response was detected, suggesting a synergistic interaction between the two oyster transcription factors. We suggest that *cgNeuroD* may function through binding to a nearby E-box (Fig. 4). However, further verification of the suggested synergistic interaction is necessary because of the complexity of *insulin* transcriptional regulation and the limitation of the heterologous cell system used in this study. We suggest there will be differences to vertebrates, because we did not detect other potential regulatory motifs nearby, in contrast to the density of such elements in vertebrates[54]. We also stress that the open chromatin peak identified in this study reflect only one chromatin regulatory status; it is possible that in some cell types there are additional conserved or novel TFBSs upstream to or neighbouring *cgILP*.

The ATAC-seq studies performed here reveal a large number of open chromatin regions close to other genes that are worthy of further investigation, including a genome-wide accessibility map for a consensus set of 584 transcription factors, providing clues to additional transcription factors potentially involved in regulating gene expression in the oyster hepatopancreas. Curiously, although we observe a larger number of putative distal cis-regulatory elements (enhancers) than of putative proximal (promoter) elements, enrichment of TFBS was primarily observed in the proximal elements, indicative of a more compact gene regulatory architecture than previously described in vertebrate systems.

Three genes containing a Pdx-binding motif in proximal ATAC-seq peaks were further assayed to test for transcription factor binding in vivo. Of these, elements upstream of *cgILP* and *cgPCSK1* both showed significant DNA enrichment after immunoprecipitation with cgPdx antibodies; a weak ATAC-seq peak upstream of *cgXBP1* did not (Supplementary Fig. 8). However, the evidence for protein-binding in vivo should be treated as indicative rather than conclusive until the newly generated antibodies used in this study are further validated[55]. For example, although western blots using E9112 and E9113 showed a signal around the expected cgPdx molecular weight, other bands are also detected, especially with E9113.

In summary, our data indicate that the Pacific oyster, a lophotrochozoan, has retained several insulin-related genes, including a putative orthologue of *insulin/IGF* genes. The *cgILP*, is co-expressed with the *cgPdx* homeobox gene and *cgPCSK1* proprotein convertase gene in the endoderm of oyster hepatopancreas, a tissue we find has gene expression similarity to the exocrine and endocrine pancreas. Experimental analyses indicate the potential ability of oyster cgPdx and cgNeuroD to activate synergistically transcription of *cgILP* through a predicted binding site, comparable to Pdx1 and NeuroD1 activation of *insulin* in vertebrates. These findings suggest that a classic homeodomain-target gene interaction dates back to the base of Bilateria. Finding such conserved interactions has important implications for comparative biology, since it allows us to move beyond comparing gene expression patterns and commence examination of the conservation of regulatory modules. The latter are likely to prove more reliable markers of evolutionary homology and enable deeper insight into evolutionary mechanisms.

## Methods

**Phylogenetic analysis and gene structure prediction**. Deduced peptide sequences for Insulin superfamily members included 20 from Mollusca, 30 from Ecdysozoa, and 51 from Deuterostomia (Supplementary Data 1). Putative orthologues between human and oyster were identified by reciprocal best-hit Blast with an $E$-value cutoff of $1e-5$[56]. Pdx sequences were retrieved from NCBI (Supplementary Fig. 3). Multiple alignment was conducted with MAFFT 7.221[57] using the L-INS-I algorithm, and trimmed with TrimAl using parameters –gt 0.9 –st 0.001 –cons 40[58]. Phylogenetic trees were constructed with RAxML[59] using the evolutionary model LG + Gamma+Invariant and 1000 bootstrap replicates. UGENE[60] was used for alignment visualisation. For oyster ILP sequences[23], signal peptides were predicted by SignalP 4.1[61], mature peptide chains predicted from adjacent pairs of basic residues[62]. For Pdx alignment visualisation, amino acids colours were coded according to their degree of evolutionary conservation and physicochemical properties using the colour scheme Zappo from Jalview[63].

**Animal material**. Three-year old Pacific oysters (*C. gigas*) used for qPCR and in situ hybridisation, RNA-seq, ATAC-seq and gene cloning were from the Oxford Covered Market, reportedly collected from the Scottish coast, and were acclimated in artificial seawater at 16 °C for 7 days before use. Oysters were fed with *Spirulina* powder during culture. Species identification was confirmed by sequencing the mitochondrial cytochrome oxidase I gene. Tissues from three individuals were dissected to obtain labial palps, gill, mantle, white adductor muscle, transparent adductor muscle, neuron, heart, haemolymph, male gonad, female gonad, hepatopancreas, stomach and intestine for immediate RNA extraction.

**RNA extraction and qPCR**. Total RNA was extracted following a standard TRIzol protocol (Invitrogen, USA) and treated with DNase I (Promega, USA). RNAs from three individuals were equally mixed before synthesis of the first-strand cDNA with GoScript™ RT (Promega, USA) according to the supplier's instructions. Primers for qPCR on oyster *insulin-like* genes, *Pdx* and as an internal control *cgEF1a* are given in Supplementary Data 4. Realtime qPCR was conducted on the Prime Pro 48 realtime qPCR system (Techne, UK). The $2^{-\Delta\Delta t}$ method was used to calculate the relative quantification (RQ) of target genes. R software[64] was used for statistics and plotting.

**Assay for transposase accessible chromatin**. To maintain osmolarity, modified Dulbecco's phosphate buffered saline (MDPBS) was used to incubate oyster cells throughout. Fresh hepatopancreas tissue fragments were incubated in 0.8 mg/ml collagenase P (Sigma, 11213857001) at room temperature for 20 min and gently pipetted every 5 min to release single cells. The cell suspension was filtered through a 40 μm cell strainer (BD Falcon, USA), transferred to a new tube, and centrifuged for 5 min at $500 \times g$. The supernatant was discarded, and the pellet suspended in MDPBS for cell viability analysis and cell counting. For each reaction 50,000 cells were collected at $1000 \times g$ for 5 min, 4 °C. The transposition reaction, purification, library construction and quantitation followed established methods[65], including tagmentation using a Nextera DNA kit (Illumina FC-121-1030) for 30 minutes at 37 °C, amplification of tagmented DNA using NEB Next High-Fidelity 2X PCR Master Mix for 11 cycles. Library quality was assessed using Agilent Tapestation.

Two samples were sequenced using 40 bp paired-end reads on the Illumina NextSeq 500 platform giving 43 and 17.5 million pairs of reads, respectively. Reads were mapped using Bowtie 2 (v. 2.1.0) to *C. gigas* genome v.9 obtained from NCBI yielding 79.8% and 74.8% of sequenced reads respectively mapped to the nuclear genome. Peaks were called on each sample using MACS2 (v. 2.1.1) using '—nomodel—shiftsize 250—nolambda' parameters. Only peaks recovered in both replicates and longer than 100 bp were retained. Peaks were further filtered to remove regions overlapping with repetitive sequences derived from a combined repeat annotation modeller (RepeatModeler[66], DUST[67] and TRF[68] filters), yielding a total of 168,206 consensus open chromatin regions genome-wide. Consensus peaks were assigned to genes denoted by *C. gigas* gene models obtained from NCBI using Homer (v.4.9) annotatePeaks.pl script. From this annotation, open chromatin regions corresponding to proximal and distal putative cis-regulatory elements were inferred. TFBS analysis was performed using the Gimme motifs suite (v.0.11.1)[69]. Position Weighted Matrices (PWMs) for individual TFBSs corresponding to PDX1 (human PDX1 sites: M5712_1.02, M5713_1.02, M6415_1.02 and mouse Pdx1 sites: M0976_1.02, M1952_1.02) were obtained from CIS-BP online library of transcription factors and their DNA binding motifs at http://cisbp.ccbr.utoronto.ca [70]. Mouse and human PWMs were clustered using gimme cluster command to yield one averaged cluster motif (consensus site) 'Pdx_Average_2' that resulted from clustering of 3 sites (Human M5713_1.02, Mouse M1952_1.02, Human M6415_1.02), a second cluster consisting of a single motif that corresponds to a doubled site of two TAAT motifs next to each other (Human M5712_1.02) and a third single motif corresponding to less prominent TAAT site (Mouse M0976_1.02). Background regions were generated using 'gimme background -random' command using *C. gigas* genome as reference. Threshold values were obtained based on background sequences using gimme threshold command with 'fpr = 0.01' for each of the individual PWMs and the Pdx_Average_2 TFBS. TFBS were identified using gimme scan using cutoff values obtained from 'gimme threshold' step on all consensus peaks. Both Pdx_Average_2 and Pdx_M1952_1.01 PWMs identified the experimentally validated Pdx1 sites

located at −1806 (TTCTAATTAC) on scaffold737:266144-266153 but failed to recognise the flanking bases (5′-T and C-3′ respectively). In addition, a pre-clustered set of 623 motifs generated from CIS-BP motifs (http://dx.doi.org/https://doi.org/10.6084/m9.figshare.1555851)[71], TFBS sites (v.3), was used to scan consensus peak set as described above. Specific Pdx1 sites were not recovered by any other clustered Homeodomain PWMs confirming specificity of analysis.

**RNA sequencing and differential expression analysis**. RNA sequencing (RNAseq) was performed using 75 bp paired-end (PE) sequencing on the Illumina HiSeq 4000 platform (Illumina, Inc., San Diego, CA, USA) from a hepatopancreas TruSeq library from the same individual as used for ATAC-seq. A total of 33.11 million PE reads were obtained with 4880 Mb Q20 yield. Data were analysed alongside published sequencing reads from additional organs (including 'digestive gland', potentially the same as hepatopancreas)[23]. Raw sequencing reads were mapped to the oyster genome with HISAT2 and further processed with the tools StringTie, and Ballgown[72]. Mapped read counts were used to identify differentially expressed genes using edgeR[73], and gene expression levels quantified as fragments per kilobase per million mapped reads (FPKM). Hepatopancreas-enriched genes were defined as those genes having higher expression levels ($p < 0.05$) in both 'digestive gland' and the newly collected 'hepatopancreas' sample compared to other organs (adductor muscle, labial palps, gill, haemolymph, mantle, male gonad, and female gonad).

**Plasmid constructs**. For riboprobe generation, amplified fragments of *cgPdx* (489 bp), *cgMIP123* (608 bp), *cgMIP4* (679 bp), *cgMILP7* (359 bp) and *cgILP* (332 bp) were cloned into pGEM-T easy (Promega, USA). For assessing subcellular localisation, *cgPdx* coding sequence was amplified with primers containing *XhoI* sites and ligated into *XhoI*-digested pEGFP-N1 vector (Clontech, USA) to generate an in-frame fusion with EGFP (construct pEGFP-Pdx). For luciferase assays, untagged *mPdx1*, *cgPdx* (LOC105327390[74]) and *cgNeuroD* (LOC105336755[75]) plasmids were constructed by cloning into *XhoI* and *MluI* sites in the pSI expression vector (Promega, USA; construct pSI-mPdx1, pSI-cgPdx, pSI-cgNeuroD). As firefly luciferase reporter constructs, corresponding genomic regions upstream of *cgILP* were ligated into the pGL4.23[*luc2*/minP] vector (Promega, USA; constructs pGL-B1). Three short versions (two with altered TAAT sites) were generated by in vitro mutagenesis (constructs pGL-B2, pGL-B2M1 and pGL-B2M2, Fig. 4). Altered TAAT sites were generated by designing mismatched base in primers covering the sites. Primers were given in Supplementary Data 4.

**Antibody synthesis and ChIP-qPCR**. The coding sequence of cgPdx peptide 1–160 was cloned into pET-28a-SUMO vector, and expressed in BL21(DE3) cells. The SUMO-tagged recombinant protein was purified by Ni-NTA affinity chromatography (Qiagen); rabbit anti-cgPdx polyclonal antibodies, E9112 and E9113, were developed by Abclonal (Wuhan, China). Validation of antibodies was conducted by western blotting on tissue lysates of oyster hepatopancreas (Supplementary Fig. 11). The full coding sequence of cgPdx was cloned in-frame (see Supplementary Data 4 for primers) with a V5 tag at the C-terminal (pSF-CMV-Puro-COOH-V5, Oxford Genetics) and expressed in HEK293T cells. Western blotting was then performed with the cell lysate to determine the molecular weight of cgPdx. Specificity of antibodies was tested by western blotting on oyster hepatopancreas lysate; anti-PARP1 antibody (ab32138, abcam, Shanghai, China) was used as a positive control. ChIP assays were performed according to a standard protocol (Abcam) with minor modifications. In brief, oyster hepatopancreas tissue pieces were dissected, frozen on liquid nitrogen and stored at −80 °C until use. Single cells were isolated using methods described for ATAC. Harvested cells were cross-linked with 1% formaldehyde for 10 min at room temperature, and soluble chromatin obtained after sonication was incubated with anti-cgPdx antibodies. Two ChIP experiments were conducted with different individuals. ~150 mg hepatopancreas tissue was used with 10 μg antibodies. Protein A (Millipore, 16–661) was used to precipitate chromatin immunocomplexes, which were washed and then DNA purified using QIAquick PCR Purification Kit (Qiagen). Primers (Supplementary Data 4) were designed for qPCR examination targeting potential Pdx-binding sites upstream three genes. ChIP DNA was then used as the template for qPCR analysis, with results normalised by input DNA. Fold enrichment was calculated according to the common 'signal over background' method (ThermoFisher). The significance of the mean of target's fold enrichment to 1 was calculated with one-sample $t$-test (one-sided).

**In situ hybridisation**. Digoxygenin-labelled probes were synthesised from linearised constructs in sense and antisense directions, used in parallel for all experiments. Whole soft body tissue from juvenile oysters (shell length 1–3 cm) was fixed in 4% paraformaldehyde in PBS overnight at 4 °C. Tissues were washed twice in PBS, and dehydrated through a graded alcohol series before storage in 70% ethanol at −20 °C. For wax embedding, tissues were washed in 85% ethanol (15 min), 100% ethanol (3 × 15 min), methyl benzoate (3 × 15 min) and benzene (1 min) before transferring into melted paraffin (3 × 30 min). Blocks were solidified at 4 °C for at least 2 h, trimmed, mounted and stored overnight (4 °C) before sectioning at 10 μm. Sections were then de-waxed with xylene and treated with proteinase K (1 μg/ml, 15 min) before hybridisation (50% deionised formamide, 5× saline sodium citrate, 0.1% Tween-20, 10% sodium dodecyl sulfate, 0.3 mg/ml torula RNA, 50 μg/

ml heparin) with 50 ng/µl probe at 65 °C, overnight. After washing with 0.2× saline sodium citrate and maleic acid buffer, riboprobes were detected by incubation with alkaline phosphatase-conjugated anti-DIG Fab fragments (1:4000; Roche Diagnostics, USA) in Roche blocking buffer in 100 mM maleic acid (2 h, room temperature). Colour development was conducted by incubation in NBT/BCIP (US Biological, Swampscott, MA). Sections were washed with distilled water, mounted in glycerol, and examined using a Leica MZ10F light microscope courtesy of Microscope Services Ltd, Oxford.

**Luciferase assay.** HeLa (ATCC, USA) and COS7 (kindly provided by Dr Qingfeng Yan, Zhejiang University, Zhejiang, China) cells were cultured in modified RPMI-1640 medium (HyClone, USA) supplemented with 10% heat-inactivated fetal bovine serum (Gibco, USA) and penicillin-streptomycin (Solarbio, China), in a humidified atmosphere with 5% $CO_2$ at 37 °C. Plasmids were transfected into cells using Lipofectamine 3000 reagent (Life Technologies, USA) according to the supplier's instructions. Luciferase assays were conducted in 24-well plates with ~250,000 cells and 500 µl medium/well; A total of 0.5 µg pSI expression plasmid DNA (by adding the blank pSI vectors to keep constant for each transfection) was transfected with 0.2 µg pGL-series reporter plasmids, and 1 ng control pRL-CMV Renilla luciferase plasmid (Promega, USA). At 24 h post-transfection, firefly and Renilla luciferase activities were measured using a Dual-Luciferase Reporter Assay System (Promega, USA) in a Varioskan Flash multimode reader (Thermo Scientific, USA). Each transfection was performed in triplicate for biological replicates, and wells divided into two to generate technical replicates. Student's $t$ test (two-sided) was used to determine the significance of differences. 95% confidence interval (CI) of the sum between cgNeuroD and cgPdx/mPdx1 was estimated based on the pool of all the possible combinations between the relative luciferase values from cgNeuroD and cgPdx/mPdx1 groups. Significance was then calculated by bootstrapping ($n = 100,000$), in which sum values with the same sample size as the group (cgPdx/mPdx1+cgNeuroD) were randomly sampled from the sum pool.

**Reporting summary.** Further information on research design is available in the Nature Research Reporting Summary linked to this article.

## Data availability
The raw sequencing data and analysis results for *C. gigas* hepatopancreas RNAseq and ATAC-seq have been deposited in the Gene Expression Omnibus under accession code: GSE107713 and under BioProject PRJNA417263. Mapping details of reads onto the genome and ATAC-seq peak calling result can also be visualised by adding the hub http://zoo-animalia.zoo.ox.ac.uk/Cragig/hub2.txt into UCSC genome browser. The authors declare that the main data supporting the findings of this study are available within the article and its Supplementary Information. Source data are provided with this paper.

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

## Acknowledgements

The authors thank Jordi Paps and Yuuri Yasuoka for discussions, Ruth Williams, Upeka Senanayake, Adam Hargreaves, Shan Quah, Ignacio Maeso, Thomas Dunwell and Qingfeng Yan for advice on ATAC-seq and cell culture, and Alan Todd for help with microscopy. The High-Throughput Genomics Group at the Wellcome Trust Centre for Human Genetics (Wellcome Trust grant 090532/Z/09/Z) performed RNA-seq. Gene expression analyses were performed using the supercomputer cluster of the High Performance Computing Center (HPCC) at the Institute of Oceanology, Chinese Academy of Sciences. F.X. acknowledges the National Natural Science Foundation of China (No. 41776152), Key Deployment Project of Centre for Ocean Mega-Research of Science, Chinese Academy of Science (COMS2019Q11) and China Scholarship Council (CSC) for financial support; P.W.H.H. acknowledges funding from the European Research Council under the European Union's Seventh Framework Programme (FP7/2007-2013 ERC grant 268513), a Royal Society International Exchanges grant and the QR GCRF award to the University of Oxford.

## Author contributions

F.X. and P.W.H.H. conceived and designed the study. F.X. conducted experiments and data analysis. F.X., F.M., D.G. and T.S.-S. conducted ATAC-seq experiments and data analysis. X.L. and G.Z. contributed to oyster experiments and analysis. F.X. and P.W.H. H. wrote the paper. All authors approved the final manuscript.

## Competing interests

The authors declare no competing interests.
