## [Peer Review File · Nature Communications]

Reviewers' Comments:

Reviewer #1:

Remarks to the Author:

The work by Xu et al entitled 'Pdx regulation of oyster insulin reveals conservation of a classic homeodomain target' deals with the characterization of an insulin-like peptide in a distant lophotrochozoan model and the putative conservation of its transcription regulation network taking the example of Pdx, a homeodomain transcription factor, described as a conserved activator of insulin transcription in oyster pancreas. The main finding of this study according to the authors is the actual conservation of the transcriptional activation of an oyster insulin gene by an oyster Pdx orthologue and the consequent evo-devo implications.

The topic of the putative conservation of a transcription module across taxa is clearly of interest for evolution and development biologists. The ATAC-seq results presented are innovative in a lophotrochozoan model. Overall, the methodology used to address the question is sound, the paper is well written and the rationale is extensively described. However, in my opinion, the manuscript presents a number of concerns regarding different aspects that could be taken into account to improve the manuscript before it may become suitable for publication. For example, it is not clear what is the real novelty of the paper in some places. The main finding of a conserved transcription module during evolution is interesting but not really surprising because it has been suspected from legions of evo-devo studies in the past decade, despite clear demonstrations of the precise underlying molecular mechanisms are lacking and would constitute a significant contribution. However, an important issue here is that the binding of Pdx to the 5' region of the oyster insulin gene is not demonstrated. I agree that the results presented (regardless the remarks below on this point) support the authors' hypothesis in a nicely presented body of corroborating indications, however, they do not constitute in my opinion a proof of what is claimed, making this study mostly associative. In addition, there are to me some methodological/interpretation shortcomings that fail to support the findings from the presented results. Some of those concerns are randomly and non exhaustively listed below.

First, I was especially surprised of the very little description and exploitation of former work on mollusk insulin-like peptides. I was pretty confused to read that the authors claim to discover 4 insulin-related peptide coding genes in the oyster (which seems to me a significant finding in itself) while they did not present any comparison with the already characterized/suspected oyster insulins thereby barely paying credit to the existing literature (Zhang et al, 2012: annotation of several ILPs in the oyster genome, Hamano 2005, cloning of an oyster insulin cDNA...). In addition, I was unable to find the accession numbers of the described oyster ILPs in the text body, but the accession numbers inferred from the sequence names in figure 1 lead to sequences already identified as oyster ILPs since February 2017 (ex:

<https://www.ncbi.nlm.nih.gov/protein/GI:762143804>). These omissions are somehow confusing.

To the same extent, the authors establish a transcriptome of oyster pancreas, but as they state such a transcriptome is already available and was co-analyzed with their own RNAseq results. I don't really see the point here of re-performing published experiments, and then melt the results in an analysis/presentation that makes unclear which is the real contribution of the authors.

Similar remarks arise from RT-qPCR results that are not compared with existing RNAseq quantifications of annotated oyster 'ILP' sequences. Besides, regarding the manuscript organization, many elements that should be found in the discussion lie in the results section, see for example the lines 96-109 on invertebrate insulins. Therefore, overall, the redaction decisions of the authors somehow 'blur the lines' between presenting/emphasizing novel contributions and existing work in some places, which seems to me fairly awkward.

There is a lack of precision in the methodology and figure presentation. For example, some controls are lacking, especially in the presentation of in-situ and IHC experiments. This is important because the oyster hepatopancreas/digestive gland is well-known to exhibit strong unspecific labelling. Therefore the appropriate controls (sense riboprobes, antibody controls) must be presented. The ATAC-seq has been done on only one biological replicate. As described, the purification of oyster pancreatic cells does not seem to allow the assessment of the purity of the cell types harvested, which is not discussed. The mutagenesis/transfection/luciferase assays are

sound, but the authors interpret an upmost-28% variation in luciferase activity, which is rather low, as an evidence of binding of Pdx to an enhancer of oyster ILP4 gene and even though (i) synergic effects of different factors are discussed, and (ii) ATAC-seq results, albeit on a single individual, are consistent with the hypothesis, it is not discussed why in oyster stomach ILP4 expression is comparable to the pancreas in absence of little Pdx expression and why in oyster intestine the Pdx expression is strong but not insulin. Could there be that the interplay between Pdx and Insulin is not that potent as hypothesized? This should be developed, especially regarding the conservation of development pathways and underlying transcription networks. Considering this, overall the study as is is 'only' (the amount of work is considerable and many aspects are well-conducted) an associative study, indeed the proof of binding of Pdx to CgILP4 enhancer is not brought. Why? A suggestion would be to perform EMSA using an anti Pdx antibody or to assay the binding in vitro for example using recombinant tagged oyster Pdx.

In the end, the contributions of the present study are real (ATAC-seq notably) but they do not provide the undoubtful evidence of the authors claims. Different aspects of the manuscript raise questions at various levels, some of which being confusing to me, and a strong effort should be done on manuscript organization and regarding the presentation of novelty and contributions. Therefore, even though the topic is clearly of evo-devo interest, in my opinion the results need to be re-presented in a new, repositioned and profoundly clarified manuscript before being reconsidered for publication.

Reviewer #2:

Remarks to the Author:

While appreciating that dealing with oyster tissue might not be the easiest proposal, and that the authors have attempted some difficult assays such as ATAC-seq for open chromatin, there are several areas where the data are not so strongly supportive of the conclusions, and much of the transcriptional analysis connecting Pdx to ILP is very indirect.

The in situ hybridization study localizing Pdx and ILP to the hepatopancreas is not convincing. The method chosen is "old school" with (stated in methods) NBT/BCIP, but many panels look like DAB development, and there is disturbing background. Or, an incomplete presentation of data, so that the true absence of high background signal in non-expressing tissue cannot be appreciated. Cannot in situ IF methods be used, or smFISH, HCR or other methods, to provide more convincing and supportive expression data. [Please note comment below regarding immunodetection of oyster Pdx protein.]

In order to establish the claim of the title, it would have been useful to have several other features strongly present in the data. First, I presume that the authors feel that the oyster Pdx sequence has already been published and so does not need to be shown here, but an alignment of the various domains (TAD, homeo, others) would be helpful shown in this paper directly. It is known that mammalian Pdx is a weak activator when provided alone, and needs coactivators for example in the bhlh group even to see moderate to strong activation of reporter constructs such as the heterologous cell-type and luciferase approach used by these authors. The conservation of the classic homeodomain-target interaction would be better explored by asking if there is any sort of motif clustering conservation around the main Pdx binding site, which could or not the clustering seen in vertebrate insulin gene cis-regulatory domains [i.e., spacing/order of Pdx and bhlh or other sites known from the large number of studies of the cis-regulatory motifs present in the vertebrate insulin genes]. The 19,000 or so ATAC-seq "peaks" are not well dealt with in the paper - what do they represent with respect to the Pdx motifs that are found throughout the genome? How many Pdx motifs are or are not under all of these peaks? The many other motifs searched for in the paper are also left in a sort of "orphaned" position once they are described - there does not seem to be much of an integration of this part with the rest of the paper. The reader is left feeling that this is the first time it's been done, yes, but the relationship to the rest of the paper is unclear.

There is no strong data indicating directly that there is occupancy of oyster Pdx at the main binding site suggested within the ILP4 5' upstream region, in oyster cells/tissue of an appropriate stage and type. Many "functional connection" experiments are done in a very ectopic context, using for example highly heterologous cell types, overexpressed proteins, and so forth. Thus there is only the circular and very indirect argument for direct interaction at the motif. Proteins overexpressed at high levels that have homeodomains and other features conserved with the vertebrate protein will very likely have no chance other than to move to the nucleus, so those experiments are not very informative. And, when such overproduced proteins are presented with luciferase reporter constructs driven by cis-regulatory regions containing embedded binding motifs, there will very likely be a weak to modest transcriptional upregulation. I do not see how the evidence presented herein is thus strongly suggestive of the conservation claimed in the paper's title, at least for the level expected to be suitable for Nature Communications.

I understand that the authors are presumably excited at the possible detection of this conservation, and the importance of such for understanding the occurrence of the association in important basal or pre-branch point animals and thus for an evolutionary angle. But I respectfully conclude that the functional data are not direct enough as of yet. Although asking quite a bit of the authors, I realize, there could be a huge amount of increased support by differently presenting the motif searching parts of the paper, and also creating an antibody that would provide clearer tissue/cell-type specificity of expression, and possibly allow chromatin IP to address the direct binding.

Reviewer #3:

Remarks to the Author:

The manuscript by Xu et al reports the very interesting finding of an evolutionary conserved regulatory link between pdx and insulin genes in considerably distant metazoans, as it is the Pacific oyster, a member of the Lophotrochozoa, compared to the vertebrates. Using transcriptome and ATAC-Seq analyses in the Pacific oyster, together with functional analysis in a heterologous system, the authors find evidence of ability of the oyster Pdx homeodomain protein to activate gene expression through a non-coding element upstream of an oyster insulin like peptide, putative orthologue of insulin.

The work is well conceived, the experiments are well done and the results are convincing. The major limitation of the paper lies in the fact that functional analysis is performed by cell culture experiments in a heterologous system and not in the species where evolutionary questions are addressed. Although I completely understand the choice of the authors of using the Pacific oyster as model system to study the evolution of a parahox gene, Pdx, which was secondarily lost in arthropods and nematodes, thing which excludes the possibility of using *Drosophila* and *Caenorhabditis* models to address these questions, I wonder if any attempt was made to experimentally test oyster Pdx function using gene editing approaches in oyster. This would greatly reinforce the finding of this paper.

This said, I appreciate that, conscious of these limitations, the authors never overstate their conclusions and elaborate an interesting critical discussion of the data.

minor points:

-line 182. How to distinguish between TFBs and consensus site? What is a consensus site in this case? The authors refer to fig S5, but it is not clear from that figure which sequences they actually used to interrogate their data

-line 187. Again, what's the difference between consensus and individual homeodomain motifs?

- line 189. Can they make clearer what they mean for "specificity" of the analysis?
- line 217. The authors do not show either in the next or in the methods how they performed site specific mutagenesis.
- line 270. The sentence need to be fixed because to my knowledge it is not possible by qPCR to check "co-expression" between two genes (Pdx1 and ILP4 in this case)
- figure 2. To a reader not familiar with ISH on oyster tissues, some data are difficult to interpret. Moreover, some signals are very faint and hard to see and to discriminate from the "no signal", as in the case of ILP4 in "j" and Pdx expression in "n". Also, it seems that they show arrows not for every site of gene expression as in "b", "j"
- figure 4, top part. The bar peaks do not correspond to the peak profile (some bars are under peaks that are very low, some high peaks do not have bars under them, some high peaks do not have bars over the all length of the peak).

Reviewer #1 (Remarks to the Author):

The work by Xu et al entitled 'Pdx regulation of oyster insulin reveals conservation of a classic homeodomain target' deals with the characterization of an insulin-like peptide in a distant lophotrochozoan model and the putative conservation of its transcription regulation network taking the example of Pdx, a homeodomain transcription factor, described as a conserved activator of insulin transcription in oyster pancreas. The main finding of this study according to the authors is the actual conservation of the transcriptional activation of an oyster insulin gene by an oyster Pdx orthologue and the consequent evo-devo implications.

The topic of the putative conservation of a transcription module across taxa is clearly of interest for evolution and development biologists. The ATAC-seq results presented are innovative in a lophotrochozoan model. Overall, the methodology used to address the question is sound, the paper is well written and the rationale is extensively described.

However, in my opinion, the manuscript presents a number of concerns regarding different aspects that could be taken into account to improve the manuscript before it may become suitable for publication. For example, it is not clear what is the real novelty of the paper in some places. The main finding of a conserved transcription module during evolution is interesting but not really surprising because it has been suspected from legions of evo-devo studies in the past decade, despite clear demonstrations of the precise underlying molecular mechanisms are lacking and would constitute a significant contribution.

However, an important issue here is that the binding of Pdx to the 5' region of the oyster insulin gene is not demonstrated. I agree that the results presented (regardless the remarks below on this point) support the authors' hypothesis in a nicely presented body of corroborating indications, however, they do not constitute in my opinion a proof of what is claimed, making this study mostly associative.

We thank the reviewer for the perceptive and helpful comments. The novelty lies in the fact that, for the most part, evo-devo studies have not demonstrated conservation of downstream targets or mechanism. As the reviewer says, downstream conservation is 'suspected' in almost every case, but we feel suspicion is not enough; the novelty here is that we try to move beyond 'suspicion' to mechanism. The second point, about whether oyster Pdx binds to the 5' region of oyster ILP, is an important question and we have now spent many additional months addressing this key point. The initial version of paper already had strong associative data (co-expression and in vitro binding and transcriptional activation), but in this revised version we have sought a more direct test. We have now raised two antibodies against oyster Pdx, and used these to ask if oyster Pdx is bound to DNA upstream of the oyster ILP4 gene. We conduct ChIP-qPCR assays and these provide direct support for binding of oyster Pdx protein to the 5' region of oyster ILP4 gene. Both antibodies of cgPdx revealed higher fold enrichment (8.86 and 9.15 times) at the predicted binding site around the ATAC-seq peak compared to the adjacent site.

In addition, there are to me some methodological/interpretation shortcomings that fail to support the findings from the presented results. Some of those concerns are randomly and non

exhaustively listed below.

First, I was especially surprised of the very little description and exploitation of former work on mollusk insulin-like peptides. I was pretty confused to read that the authors claim to discover 4 insulin-related peptide coding genes in the oyster (which seems to me a significant finding in itself) while they did not present any comparison with the already characterized/suspected oyster insulins thereby barely paying credit to the existing literature (Zhang et al, 2012: annotation of several ILPs in the oyster genome, Hamano 2005, cloning of an oyster insulin cDNA...).

We were perhaps not clear enough in our description of background data; the Zhang et al paper is actually our work. We had previously discovered the 4 insulin-related genes, but this earlier work (and that of others) did not include detailed co-expression work nor protein binding analyses and the other evolutionary analyses in the current paper. It is these that underpin the more significant finding. We now added one extra paragraph (lines 271-291) to discuss previous studies on mollusk insulin-like peptides, including our and others' reports of oyster ILPs,

In addition, I was unable to find the accession numbers of the described oyster ILPs in the text body, but the accession numbers inferred from the sequence names in figure 1 lead to sequences already identified as oyster ILPs since February 2017 (ex: <https://www.ncbi.nlm.nih.gov/protein/GI:762143804>). These omissions are somehow confusing.

We apologize for any confusion. We have now corrected the sentence to "Four insulin-like genes were identified previously in the Pacific oyster genome²²".

To the same extent, the authors establish a transcriptome of oyster pancreas, but as they state such a transcriptome is already available and was co-analyzed with their own RNAseq results. I don't really see the point here of re-performing published experiments, and then melt the results in an analysis/presentation that makes unclear which is the real contribution of the authors.

There was a strong reason for collecting new data. The earlier digestive gland (Dgl) transcriptome data in NCBI was also produced by the first author of the current study. However, that sample was less precise in tissue type, and encompassed a wider region of gut, plus possibly some gonad. In this study, we generated new RNAseq data precisely on hepatopancreas (Hep) by removing stomach and intestine. An explanation has been added in lines 145-146.

Similar remarks arise from RT-qPCR result that are not compared with existing RNAseq quantifications of annotated oyster 'ILP' sequences. Besides, regarding the manuscript organization, many elements that should be found in the discussion lie in the results section, see for example the lines 96-109 on invertebrate insulins. Therefore, overall, the redaction decisions of the authors somehow 'blur the lines' between presenting/emphasizing novel contributions and existing work in some places, which seems to me fairly awkward.

The reason for focusing on the new RNAseq data is due to precision of dissection, as explained above. In addition, a second novel contribution is to deploy RT-qPCR to determine gene expression levels in neuron, stomach and intestine. These are key tissues to study for insulin-like

genes, but missed in previous RNAseq data. Concerning the split between Results and Discussion, previous lines 96-109 on invertebrate insulins have been moved to discussion part as suggested. All ILP sequences were identified by previous report, but phylogenetic analysis and peptides structure analysis were novel contributions in this study. This has now been clarified.

There is a lack of precision in the methodology and figure presentation. For example, some controls are lacking, especially in the presentation of in-situ and IHC experiments. This is important because the oyster hepatopancreas/digestive gland is well-known to exhibit strong unspecific labelling. Therefore the appropriate controls (sense riboprobes, antibody controls) must be presented.

As explained in Methods, sense and antisense riboprobes were used in parallel for all experiments, providing appropriate controls. We now show control data in a revised figure 2.

The ATAC-seq has been done on only one biological replicate. As described, the purification of oyster pancreatic cells does not seem to allow the assessment of the purity of the cell types harvested, which is not discussed.

As we indicated in the Materials and Methods section, we actually generated two replicates of ATAC-seq data, both of which have been submitted to GEO under accession number GSE107713. Only the peaks that were recovered in both replicates were used for analyses. We have now made this point clearer in the manuscript. The goal of the experiment was to characterise the open chromatin regions in the hepatopancreas, disregarding possible differences between cell types making up the organ as a whole. We carefully dissected the hepatopancreas and dissociated cells according to standard protocols that should have preserved the integrity of all cell types within the hepatopancreas. We carefully examined dissociated cells and nuclei under the microscope and confirmed the absence of cell clumps or debris. Characterisation of open chromatin profiles in each cell type would require either a transgenic oyster line permitting FACS-sorting based on cell-type specific markers single-cell analysis, both of which are rather complex experiments and beyond of the scope of this paper as we were interested in detecting the open chromatin regions at the organ level rather than the cell type level.

The mutagenesis/transfection/luciferase assays are sound, but the authors interpret an upmost-28% variation in luciferase activity, which is rather low, as an evidence of binding of Pdx to an enhancer of oyster ILP4 gene and even though (i) synergic effects of different factors are discussed, and (ii) ATAC-seq results, albeit on a single individual, are consistent with the hypothesis, it is not discussed why in oyster stomach ILP4 expression is comparable to the pancreas in absence of little Pdx expression and why in oyster intestine the Pdx expression is strong but not insulin. Could there be that the interplay between Pdx and Insulin is not that potent as hypothesized? This should be developed, especially regarding the conservation of development pathways and underlying transcription networks. Considering this, overall the study as is is 'only' (the amount of work is considerable and many aspects are well-conducted) an associative study, indeed the proof of binding of Pdx to CgILP4 enhancer is not brought. Why? A suggestion would be to perform EMSA using an anti Pdx antibody or to assay the binding in vitro

for example using recombinant tagged oyster Pdx.

We agree that the association between expression level of transcription factor and putative target gene are not precise, but there are many possible reasons for this, such as co-factors as discussed. As requested, we have clarified this discussion in the revised version. In terms of additional support requested, we think that another in vitro test (such as EMSA with a tagged protein) would not provide any stronger evidence that the in vitro luciferase test already presented. Instead, we felt it was worth investing time, cost and energy in an in vivo test of binding. Therefore, as explained above, we have now raised antibodies against oyster Pdx protein and conducted chromatin immunoprecipitation.

In the end, the contributions of the present study are real (ATAC-seq notably) but they do not provide the undoubtful evidence of the authors claims. Different aspects of the manuscript raise questions at various levels, some of which being confusing to me, and a strong effort should be done on manuscript organization and regarding the presentation of novelty and contributions. Therefore, even though the topic is clearly of evo-devo interest, in my opinion the results need to be re-presented in a new, repositioned and profoundly clarified manuscript before being reconsidered for publication.

Definitive proof of biochemical mechanisms is always going to be elusive, especially in non-model organisms, but we hope that the new data added to the paper go some way to reducing the uncertainty of the reviewer.

--

Reviewer #2 (Remarks to the Author):

While appreciating that dealing with oyster tissue might not be the easiest proposal, and that the authors have attempted some difficult assays such as ATAC-seq for open chromatin, there are several areas where the data are not so strongly supportive of the conclusions, and much of the transcriptional analysis connecting Pdx to ILP is very indirect.

The in situ hybridization study localizing Pdx and ILP to the hepatopancreas is not convincing. The method chosen is "old school" with (stated in methods) NBT/BCIP, but many panels look like DAB development, and there is disturbing background. Or, an incomplete presentation of data, so that the true absence of high background signal in non-expressing tissue cannot be appreciated. Cannot in situ IF methods be used, or smFISH, HCR or other methods, to provide more convincing and supportive expression data. [Please note comment below regarding immunodetection of oyster Pdx protein.]

We thank the reviewer for the suggestions. We have looked carefully at many repeats of these experiments and are fully convinced of the in situ data. We also note that it is consistent with RNAseq and with RT-PCR. However, we were remiss not to show the control data in the figure, so we have now added these to the figure.

In order to establish the claim of the title, it would have been useful to have several other features strongly present in the data. First, I presume that the authors feel that the oyster Pdx sequence has already been published and so does not need to be shown here, but an alignment of the various domains (TAD, homeo, others) would be helpful shown in this paper directly. It is known that mammalian Pdx is a weak activator when provided alone, and needs coactivators for example in the bhlh group even to see moderate to strong activation of reporter constructs such as the heterologous cell-type and luciferase approach used by these authors. The conservation of the classic homeodomain-target interaction would be better explored by asking if there is any sort of motif clustering conservation around the main Pdx binding site, which could or not the clustering seen in vertebrate insulin gene cis-regulatory domains [i.e., spacing/order of Pdx and bhlh or other sites known from the large number of studies of the cis-regulatory motifs present in the vertebrate insulin genes].

A sequence alignment for Pdx was provided in the supplementary figure S3 and was analyzed in the main text. There are several possible binding sites for other TFs such as forkhead factors or nuclear receptors, but perhaps not surprisingly the similarity in overall architecture is not strong to mammalian insulin genes. We have added discussion of this to the main text.

The 19,000 or so ATAC-seq "peaks" are not well dealt with in the paper - what do they represent with respect to the Pdx motifs that are found throughout the genome? How many Pdx motifs are or are not under all of these peaks? The many other motifs searched for in the paper are also left in a sort of "orphaned" position once they are described - there does not seem to be much of an integration of this part with the rest of the paper. The reader is left feeling that this is the first time it's been done, yes, but the relationship to the rest of the paper is unclear.

We did not search for Pdx sites throughout the whole genome as this is likely to yield false positives and would not be conclusive. Instead, we only searched for Pdx sites in the open chromatin regions that are likely to be utilised by the organism. We computationally randomly generated a non-overlapping set of oyster (v.9) genomic sequences of corresponding size that we used as background (and therefore not expected to be enriched in Pdx motifs). The 19,104 putative Pdx sequences correspond to the sites whose PWM matching score is above threshold value as determined by contrast against background sequences with an fpr=0.01 (false positive rate). To ascertain specificity, we furthermore, searched for 623 transcription factor binding sites in these open chromatin regions, and found that 583/623 were present in these open chromatin regions. Importantly, although homeodomain transcription factors can frequently bind similar sites, in our TFBS screen, we did not find any other predicted homeodomain transcription factor binding sites overlapping with the predicted Pdx sites. We feel that this is more convincing of the specificity of our approach than searching for TAAT-like sequences throughout the entire genome, that we would expect very large numbers of. In comparison, experiments carried out by others in human pancreatic islets have shown 18,294 ChIPseq peaks ²⁸ or ~6000 ATAC-seq peaks ²⁹, however we feel that comparisons of such numbers can be misleading. Total numbers of peaks

are heavily dependent on methods utilised for analysis and experimental procedures and are not easily comparable between datasets from the same species. But most importantly numbers of *cis*-regulatory elements cannot be transposed across large evolutionary distances between humans and oysters.

There is no strong data indicating directly that there is occupancy of oyster Pdx at the main binding site suggested within the ILP4 5' upstream region, in oyster cells/tissue of an appropriate stage and type. Many "functional connection" experiments are done in a very ectopic context, using for example highly heterologous cell types, overexpressed proteins, and so forth. Thus there is only the circular and very indirect argument for direct interaction at the motif. Proteins overexpressed at high levels that have homeodomains and other features conserved with the vertebrate protein will very likely have no chance other than to move to the nucleus, so those experiments are not very informative. And, when such overproduced proteins are presented with luciferase reporter constructs driven by *cis*-regulatory regions containing embedded binding motifs, there will very likely be a weak to modest transcriptional upregulation. I do not see how the evidence presented herein is thus strongly suggestive of the conservation claimed in the paper's title, at least for the level expected to be suitable for Nature Communications.

Clearly the lack of direct evidence of Pdx binding *in vivo* is the main concern of reviewer 1 and 2. As explained above, to address this concern we have now raised antibodies against oyster Pdx and used these to ask if oyster Pdx is present – *in vivo* - at the inferred binding site upstream of the oyster ILP4 gene. We conducted CHIP-qPCR with adult oyster hepatopancreas, and found ~10X enrichment of Pdx binding at the inferred binding site than at the adjacent region. This provide some direct support of our claim, although we acknowledge that some degree of inference is always going to be present. We should also stress that the luciferase experiments did not show up-regulation with every motif containing TAAT (e.g. GCCTAATGAT and ATCTAATTTT did not respond), again arguing that the upregulation at the TTCTAATTAC site is specific.

I understand that the authors are presumably excited at the possible detection of this conservation, and the importance of such for understanding the occurrence of the association in important basal or pre-branch point animals and thus for an evolutionary angle. But I respectfully conclude that the functional data are not direct enough as of yet. Although asking quite a bit of the authors, I realize, there could be a huge amount of increased support by differently presenting the motif searching parts of the paper, and also creating an antibody that would provide clearer tissue/cell-type specificity of expression, and possibly allow chromatin IP to address the direct binding.

The transcription factor binding search was carried out using statistical tests with stringent cutoffs , and was carried out on >600 transcription factor binding sites. If our TFBS-screening approach was non-specific we would expect more overlap, which we do not observe. Sections of the manuscript regarding motif searching have been carefully revised as suggested. Importantly, we have also now raised antibodies and confirmed bioinformatic predictions and luciferase tests via CHIP experiments.

--

Reviewer #3 (Remarks to the Author):

The manuscript by Xu et al reports the very interesting finding of an evolutionary conserved regulatory link between pdx and insulin genes in considerably distant metazoans, as it is the Pacific oyster, a member of the Lophotrochozoa, compared to the vertebrates. Using transcriptome and ATAC-Seq analyses in the Pacific oyster, together with functional analysis in a heterologous system, the authors find evidence of ability of the oyster Pdx homeodomain protein to activate gene expression through a non-coding element upstream of an oyster insulin like peptide, putative orthologue of insulin.

The work is well conceived, the experiments are well done and the results are convincing. The major limitation of the paper lies in the fact that functional analysis is performed by cell culture experiments in a heterologous system and not in the species where evolutionary questions are addressed. Although I completely understand the choice of the authors of using the Pacific oyster as model system to study the evolution of a parahox gene, Pdx, which was secondarily lost in arthropods and nematodes, thing which excludes the possibility of using *Drosophila* and *Caenorhabditis* models to address these questions, I wonder if any attempt was made to experimentally test oyster Pdx function using gene editing approaches in oyster. This would greatly reinforce the finding of this paper.

We thank for the interesting suggestion. Oyster gene editing has been reported this year, but the technique is limited by low viability and high malformation rate. Instead, we chose to raise antibodies and used these to overcome the limitations of a heterologous test: we conducted ChIP to test for direct binding.

This said, I appreciate that, conscious of these limitations, the authors never overstate their conclusions and elaborate an interesting critical discussion of the data.

minor points:

-line 182. How to distinguish between TFBs and consensus site? What is a consensus site in this case? The authors refer to fig S5, but it is not clear from that figure which sequences they actually used to interrogate their data

We apologise for not being clearer in the original manuscript. Most publically available molecular data for Pdx binding, and therefore binding sites, were generated in mouse and human. We suspected that motifs bound by the oyster Pdx protein may not be identical to those bound by Pdx protein in mouse or human. This was indeed the case as we found very few Pdx mouse sites in our dataset. Therefore, we utilised the 'gimme motif' computational pipeline to cluster the available mouse and human position weighted matrices (motifs) using the WIC metric. Using this pipeline, we were able to generate a "consensus" motif, which resulted in providing a consensus

site that incorporates the common elements of several sites bound by Pdx. This allowed us to predict the oyster Pdx-bound motifs which we then confirmed experimentally.

-line 187. Again, what's the difference between consensus and individual homeodomain motifs?

As explained above, the individual motifs are the publically available mouse or human TFBS motifs. Consensus motifs are the "average" motifs obtained by combining several mouse or human motifs to obtain an average TFBS. We have made these terms clearer throughout the paper.

- line 189. Can they make clearer what they mean for "specificity" of the analysis?

In this context, we meant that the Pdx motif was the only homeodomain one detected in the corresponding regions.

- line 217. The authors do not show either in the next or in the methods how they performed site specific mutagenesis.

Method was added "Altered TAAT sites were generated by designing mismatched base in primers covering the sites".

- line 270. The sentence need to be fixed because to my knowledge it is not possible by qPCR to check "co-expression" between two genes (Pdx1 and ILP4 in this case)

We are not sure what the concern is. qPCR on the same RNA sample can detect expression of two genes. We have changed the sentence to be more explicit: "Using qPCR and *in situ* hybridisation we found that the expression of *ILP4*, and no other oyster *ILP* gene, overlapped with the Pdx homeobox gene in the endodermal cell layer of adult oyster hepatopancreas and other gut tissues".

- figure 2. To a reader not familiar with ISH on oyster tissues, some data are difficult to interpret. Moreover, some signals are very faint and hard to see and to discriminate from the "no signal", as in the case of ILP4 in "j" and Pdx expression in "n". Also, it seems that they show arrows not for every site of gene expression as in "b", "j"

Figure 2 has been reconstructed according to the comments.

- figure 4, top part. The bar peaks do not correspond to the peak profile (some bars are under peaks that are very low, some high peaks do not have bars under them, some high peaks do not have bars over the all length of the peak).

The mapped and processed binned ATAC-seq signal is represented on top of the figure in a histogram-like fashion as visualised in UCSC genome browser in wig format. When we speak of peaks we refer to the "called peaks". The called peaks are visualised as the "bars" (bed format in

UCSC genome browser) that the reviewer refers to. The peak calling process is not a direct extrapolation of the regions with the highest signal. Instead, it takes into account the averaged background over a window as well as the shift induced by the library construction process. Furthermore, only peaks represented in both replicates were retained, which can also explain why some high-signal regions in one replicate do not result in a consensus peak. The wig and bed format data were submitted to GEO.

Reviewers' Comments:

Reviewer #1:

Remarks to the Author:

Review of the revised submission NCOMMS-18-16024A-Z – comments to authors.

The authors did respond to most of my former concerns in the revised submission. Especially, they undertook important additional work that brings new data to the community and now strongly supports their principal claim, that Pdx binds to a regulatory element in the proximal promoter upstream the oyster insulin gene they examine. Although in my opinion the transcription activation strength of the Pdx pathway is very weak in the luciferase experiments and poorly convincing in itself, the observation that it seems within the range of its transcriptional activation magnitude in other clades thereby reinforces the authors assessments of the conservation of a homeobox transcription network. The additional ChIP-qPCR experiments in the revised submission are sound and brings the missing link. It represents substantial gain that, together with the presented data, clearly demonstrates the main finding which now opens debate in a strongly reliable way. The study is now undoubtedly consistent, straightforward, supportive and robust.

However, there is still an important concern in my opinion that has not been addressed, which is quite confusing and requires revision. In the first version of the manuscript I remarked that the authors did not pay sufficient credit to existing studies on mollusc insulins. Their response is that i) they made clearer in the rebuttal letter that some authors were part of the oyster genome project, and ii) they added a paragraph in the discussion dedicated to mollusc insulins. However, there is still a study by others which is not mentioned albeit dealing with the molecular characterization, phylogenetics and expression (including RT-qPCR confronted to existing RNAseq data and in-situ hybridization) of oyster insulins seemingly including the target of Pdx described in the present paper (Cherif-Feildel et al. Gene Comp Endocrinol 2019 Molecular evolution and functional characterisation of insulin related peptides in molluscs: Contributions of Crassostrea gigas genomic and transcriptomic-wide screening and the related Data in Brief report). Although this study was already available at the time of the first submission, it was recent and might have been mistakenly not included in the original manuscript during the course of the submission. However, this is not the case anymore, and oyster ILPs were characterized, at least to some extent, by others. Authorship of the genome annotation is not equivalent to gene characterization and therefore, it is not appropriate nor possible for the authors not to mention and confront this work. Maybe this reference has not been disclosed to the author's knowledge. Or maybe there are valid reasons accounting for this situation and they have to be clearly stated. Whatever, authors must refer to this study which cannot be considered a 'side' or redundant work in the present context. They have to confront their own results with this existing literature and remove the material already published, because it is misleading in the present form. A thorough edition of the paper is needed, including as to my point of view, at least deep revision, if not removal of :

- results paragraph 1;
- parts of results paragraph 2 (qPCR data essentially, maybe all the paragraph excluding the first sentence);
- All the related parts of the discussion, ie. Paragraphs related to insulin characterization and phylogenetics;
- Methods paragraph 1;
- Figure 1;
- Figure 2;
- Figure 6;
- Supplementary table S2;
- Supplementary figure S2.

Indeed, it is easily noticeable that these parts of the manuscript, including figures, are almost identical in content -if not in form- to what is found in (Cherif-Feildel et al. 2019 Gen Comp Endocrinol) and (Cherif-Feildel et al. 2019 Data in Brief). The authors are prompted to properly cite this work and remove duplicated material. A very rigorous care has to be applied throughout the manuscript so that no ambiguity remains about existing data vs. new contribution. Although I

understand it represents some additional editing work, my opinion is that citing/removing/confronting such material will let the authors further discuss transcription networks evolution and conservation and even better focus on this main finding.

Reviewer #2:

Remarks to the Author:

I thank the authors for the serious manner in which they addressed the critiques from the reviewers.

There were, however, some serious omissions from responses to some of my specific comments. I hope that this was not due to strange wording from me, from which the authors were not able to understand my real expectation for the testing of the transcriptional activity of cgPdx that might lead to much stronger support for a "moderate" level of positive activation activity. As it is, the fold-activations are only 1.17 or thereabouts, and some Figures (5c) only go from 1.17 to 1.23 to 1.27 as the cgPdx goes up from 100 to 500 ng plasmid transfected - this is a tiny activation, and I am surprised that the error bars are so tight with such data. But, leaving that latter concern alone, I would be much more impressed by a slightly deeper test of conservation of mechanism, and tried to suggest this strongly in my first review (missed by the authors?). As the authors do or should know, the vertebrate Pdx protein is, when assayed by itself, a weak transcriptional activator of luciferase constructs that are driven by vertebrate insulin-gene cis-regulatory regions. But, when cotransfected with a bHLH factor such as NeuroD there is a much more robust response. It could be that this species does not use Pdx:(bhlh)-type cooperativity to activate its ILP4 gene, but I would be surprised. And if the authors did the experiment to test this idea, and it did not show stronger activation, I would actually find the result interesting in and of itself. Other experiments from the Montminy group focused on the cooperativity between Pdx and Pbx factors. I would suggest that the following are simple, and could be inserted to this m/s. The cgPdx alone could be compared to the vertebrate Pdx1, in both cases +/- NeuroD, and the strength of cgPdx as an activator, in the authors' own hands, could be then tested against vertebrate factors in a vertebrate cell line.

The authors claim that "we identify an upstream regulatory element of the oyster ILP4 gene directly bound by Pdx protein in oyster hepatopancreas and demonstrate, using a cell culture assay, that oyster Pdx acts as a transcriptional activator through this site." For the sake of correctness, however, they have produced evidence consistent with this idea, but not demonstrated that it DOES act through this site. There is the issue of the ChIP referred to below, but the heterologous cell-culture luc-reporter assays show that this factor CAN act here (for a minimal effect - more below), rather than it does act here - again, evidence in SUPPORT of the view that cgPdx does similar things to cgILP4 that vertebrate Pdx1 does to Insulin. This type of text needs to be modified accordingly, and checked elsewhere for this strict interpretation.

In a related vein, I find the overall title of the manuscript to be non-descriptive of what has been discovered, and recommend reworking it.

The main text states that Pdx1 binds target genes in liver - but this is not the case, and it's very misleading. The cited article (#39) is a study of reprogramming of hepatic cells by Pdx1 - not akin to normal liver at all. Pdx1 is not expressed in normal liver cells. Indeed, as well as fixing this issue, they might want to refer to the ability of Pdx1 to show repressive as well as activating activity, context-dependent, but also even in the same cell type (vertebrate mature insulin-

producing beta cells). When ESC are differentiated into pancreatic beta cells, there is a stage where Pdx1 begins to exert its repressive influence on other (e.g., liver genes - Teo et al., Stem Cell Reports, 2015).

To be explicit again, the authors should discuss in a detailed manner how little conservation there is to the relatively deeply characterized vertebrate insulin locus - how none or a very few of the motifs are present in a tightly spaced manner in the cg ILP4 cis-regulatory region surrounding their B4 site.

Figure 4 is very hard to follow - it needs some cosmetic alterations. I can't tell what's going on with the ATAC-seq landscape (ATAC-Seq R1) and the bars that are supposed to be indicating the called peaks. There are peaks that are in other exons and in other regions that are not underlain by the blue boxes - why is this - please explain in the legend. Is this correct? If so, the text of the legend needs to explain it. In the enlargement, why does the blue bar come to an end halfway through the main peak at the right hand side? Also: Purely minor cosmetic: more space in between the "* additional Pdx site" and the line underneath, bottom left text of the panel? The four exons of the ILP4 gene are indicated in the top bar? and the TSS is at the left end, transcription to the right? Please indicate.

While I clearly do understand that there are no cg cultured cell lines or transgenic approaches that would address how the cis-reg regions of IL4 are regulated by Pdx, it is, to me, worth including a "rationale" statement for the reader, where the move to the HeLa cell line is justified as a minimal albeit extremely heterologous context for the ability of cgPdx to mount an activation response.

The ChIP experiment may - and I emphasize "may" - be flawed. The antibodies were, mistakenly in my view, generated against the entire CDS of cgPdx, leading to potential problems with antibodies being generated against the homeodomain itself, which may share substantial similarity/identity with homeodomains of other proteins present in cg. Therefore the experiment in Fig. 5d might not unequivocally show Pdx occupancy at the B4 site. Agreed the test against B3 shows something binding at the B4-containing PCR amplicon region, which is good. But is it Pdx? Something must be shown about the specificity of the antibodies, and whether or not they react with homeoproteins of other types. While discussing this figure, it is also standard to have tested against non-specific IgG for the degree of enrichment of binding at this region - the result with B3 versus B4 seems good, but could the authors simply show us what happened with the IgG-alone control too, please?

I suspect this is a matter of style, and the journal itself may not care much at all, but many parts of the discussion read rather redundantly or repetitively with the text of the introduction or results sections. In some places, rather too "listy" of all that was done, and for my preference this makes the reading the discussion quite wearing.

The various tissues in the lower-magnification images of Figure 2 need labeling otherwise it's just not useful to look at. This figure needs some major cosmetic improvement to increase the use to the reader. A very few of the reader population will be oyster experts.

Minor comments:

Where "residues" are described as affecting the ability of the core TAAT to bind homeodomain factors, this should refer to nucleotides not (amino acid) residues, I believe.

Some parts of the text read awkwardly. The subtitle of the results text "Identification of non-coding regulatory elements by ATAC-seq" reads oddly, because ATAC-seq just reports open chromatin, not non-coding regulatory elements. I suggest reworking this subtitle. The description of amino acids at several key sites having "similar physicochemical properties" just seems odd again - residues are conservatively substituted? r "residues have similar R-group character"?

At the text "Curiously, although, we observe a larger number of..." should the second comma be there? Removal makes it easier to read.

At text "We hypothesized that one of these sites to be a candidate for Pdx-binding site in oyster hepatopancreas. Possible binding sites of other insulin-related transcription factors were also identified around 1,806 bp site peak, including FoxO forkhead transcription factor (at -1675 bp: CTGTTTAC) and nuclear receptors (e.g. HNF4A at -1734 bp: GAGGTCA).", the language precision has drifted a little - please check for correct grammar and readability.

Reviewer #3:

Remarks to the Author:

I know this work because I have reviewed the previous version of this manuscript. I can appreciate that the authors considerably improved the manuscript, mainly by producing an oyster PDX1 antibody and performing chromatin immunoprecipitation experiments to confirm the main point of this paper, i.e., the Pdx gene is upstream of insulin in oyster.

The current manuscript is therefore much more solid now than before and the point is clearly made. However, I still think the manuscript can improve in the presentation of the data, in particular at the figure level, and with some additional analysis. In particular:

- Fig. 1: It is not easy to identify the oyster insulin like peptides in the tree. I would suggest to highlight the names of the oyster ILP1, ILP2, ILP3 and ILP4 in bold and/or different colour to spot them at a first glance. Also, please spell out the abbreviation MIP, or at least the M of it, also to be consistent with the other group names (Amphioxus, Nematode, etc)
- Fig. 2: The current size of the panels is too small; I suggest to arrange the panels in a different way, to have a figure taller than larger, so to have less panels per row and, overall, bigger panels. If there were a size limitation from the journal, then I would reduce the number of panels, transferring some of them in a Supplementary figure
- Fig. 3: same comments as Fig. 2; for this figure, in case, due to space constraints, a reduction of panels were necessary, I would suggest to give a major space to panels (d) and (e) and move (a), (b) and (c) to Supplementary material; in addition, the title of this figure is very little informative; I would use a title which is more indicative of the real content of the figures and the results conceived by it; like it is right now "Assessment of the quality of ATAC-seq data" the figure should be transferred entirely to Supplementary material. Indeed, there is useful info there (at least panel d and e) and these should be highlighted and better visible
- Fig. 4: also here the title of the figure is very boring, actually this is a necessary description, but not a title. Please give a title to this figure which says a bit more about its content
- Fig. 5: also here I would change the title, to a less technical description. Do not know, maybe something like "Nuclear localization, transcriptional activity and chromatin immunoprecipitation of oyster Pdx protein" or even more generic, "Nuclear localization and functional characterization of oyster Pdx protein"; also, please better describe what is reported in (d)
- An additional comment here concerns the analysis of the ATAC-seq and ChIP data. The authors state that: "We scanned these accessible chromatin peaks for consensus Pdx motifs obtained through TFBS clustering of five Pdx1 Transcription Factor Binding Sites (TFBSs) derived from the literature (Supplementary Figure S6). A total of 19,104 putative Pdx binding sites were deemed statistically significant within the 168,206 accessible chromatin peaks ($P < 0.05$). The number is

comparable with observed binding peaks in human pancreatic islet analysed with ChIPseq (18,294) 28 or ATACseq (~6000) 29." This is very interesting and is a clear indication that Pdx might have a role in controlling hepatopancreas specific genes in oyster similarly to what the homologous gene does in the vertebrate pancreas. Now, since the authors have produced and analysed a good hepatopancreas transcriptome and since they also have produced oyster Pdx ChIPed material, it would be interesting to know at which level the ATAC seq and RNAseq data overlap; in other words, it would be interesting to analyse how many of the PDX binding site enriched peaks are putatively controlling hepatopancreas DE genes and/or validate a few of these targets by qPCR of the oyster Pdx ChIPed material; this additional analysis, which the authors have probably made already, in my opinion, would add strength to the finding of a conserved transcriptional role of Pdx

Responses to reviewers' comments:

Reviewer #1 (Remarks to the Author):

Review of the revised submission NCOMMS-18-16024A-Z – comments to authors.

The authors did respond to most of my former concerns in the revised submission. Especially, they undertook important additional work that brings new data to the community and now strongly supports their principal claim, that Pdx binds to a regulatory element in the proximal promoter upstream the oyster insulin gene they examine. Although in my opinion the transcription activation strength of the Pdx pathway is very weak in the luciferase experiments and poorly convincing in itself, the observation that it seems within the range of its transcriptional activation magnitude in other clades thereby reinforces the authors assessments of the conservation of a homeobox transcription network. The additional ChIP-qPCR experiments in the revised submission are sound and brings the missing link. It represents substantial gain that, together with the presented data, clearly demonstrates the main finding which now opens debate in a strongly reliable way. The study is now undoubtedly consistent, straightforward, supportive and robust.

However, there is still an important concern in my opinion that has not been addressed, which is quite confusing and requires revision. In the first version of the manuscript I remarked that the authors did not pay sufficient credit to existing studies on mollusc insulins. Their response is that i) they made clearer in the rebuttal letter that some authors were part of the oyster genome project, and ii) they added a paragraph in the discussion dedicated to mollusc insulins. However, there is still a study by others which is not mentioned albeit dealing with the molecular characterization, phylogenetics and expression (including RT-qPCR confronted to existing RNAseq data and in-situ hybridization) of oyster insulins seemingly including the target of Pdx described in the present paper (Cherif-Feildel et al. Gene Comp Endocrinol 2019 Molecular evolution and functional characterisation of insulin related peptides in molluscs: Contributions of Crassostrea gigas genomic and transcriptomic-wide screening and the related Data in Brief report). Although this study was already available at the time of the first submission, it was recent and might have been mistakenly not included in the original manuscript during the course of the submission. However, this is not the case anymore, and oyster ILPs were characterized, at least to some extent, by others. Authorship of the genome annotation is not equivalent to gene characterization and therefore, it is not appropriate nor possible for the authors not to mention and confront this work. Maybe this reference has not been disclosed to the author's knowledge. Or maybe there are valid reasons accounting for this situation and they have to be clearly stated. Whatever, authors must refer to this study which cannot be considered a 'side' or redundant work in the present context. They have to confront their own results with this existing literature and remove the material already published, because it is misleading in the present form. A thorough edition of the paper is needed, including as to my point of view, at least deep revision, if not removal of :

- results paragraph 1;

It is correct that Cherif-Feildel et al. present a very good descriptive analysis of oyster ILPs but we stress that (a) the data that we present are complementary not identical, and (b) the Cherif-Feildel et al. paper is purely descriptive whereas we add functional studies. In response to this issue, we

have shortened the descriptive part of our manuscript, to focus only on findings new to the field; we have also combined two sections from our previous manuscript ('Oyster has a diversity of insulin-related genes' and 'Co-expression of ILP4 and Pdx in oyster digestive tissue'). Possibly because we sampled a broader range of taxa to conduct the phylogenetic analysis of oyster ILPs, our interpretation of the pathway of gene duplication in the gene superfamily is different from that proposed by Cherif-Feildel et al.; we therefore retained figure 1.

- parts of results paragraph 2 (qPCR data essentially, maybe all the paragraph excluding the first sentence);

This paragraph was reduced, retaining only qPCR and in situ results that did not duplicate earlier reports; most importantly, we show co-expression of *Cg-ILP* and *cgPdx* in endodermal tissue and qPCR on dissection of oyster digestive gland, experiments of key relevance to our work and neither reported previously.

- All the related parts of the discussion, ie. Paragraphs related to insulin characterization and phylogenetics;

Deleted

- Methods paragraph 1;

We found that our wider sampling of taxa resulted in slightly different results compared to Cherif-Feildel, so we retained this part.

- Figure 1;

See above.

- Figure 2;

The figure has now been revised. In particular, we have renamed genes according to Cherif-Feildel et al 2019 to ensure consistency in the literature. Some in situ figures were removed.

- Figure 6;

Deleted

- Supplementary table S2;

We retained this table to support the phylogenetics result. Gene names are modified for consistency with Cherif-Feildel et al 2019.

- Supplementary figure S2.

We retained this table to support Figure 1. Gene names are modified for consistency with Cherif-Feildel et al 2019.

Indeed, it is easily noticeable that these parts of the manuscript, including figures, are almost identical in content -if not in form- to what is found in (Cherif-Feildel et al. 2019 Gen Comp Endocrinol) and (Cherif-Feildel et al. 2019 Data in Brief). The authors are prompted to properly cite this work and remove duplicated material. A very rigorous care has to be applied throughout the manuscript so that no ambiguity remains about existing data vs. new contribution. Although I understand it represents some additional editing work, my opinion is that citing/removing/confronting such material will let the authors further discuss transcription networks evolution and conservation and even better focus on this main finding.

Thanks for the helpful suggestions. We have carefully edited the content as suggested.

--

Reviewer #2 (Remarks to the Author):

I thank the authors for the serious manner in which they addressed the critiques from the reviewers.

There were, however, some serious omissions from responses to some of my specific comments. I hope that this was not due to strange wording from me, from which the authors were not able to understand my real expectation for the testing of the transcriptional activity of cgPdx that might lead to much stronger support for a "moderate" level of positive activation activity. As it is, the fold-activations are only 1.17 or thereabouts, and some Figures (5c) only go from 1.17 to 1.23 to 1.27 as the cgPdx goes up from 100 to 500 ng plasmid transfected - this is a tiny activation, and I am surprised that the error bars are so tight with such data. But, leaving that latter concern alone, I would be much more impressed by a slightly deeper test of conservation of mechanism, and tried to suggest this strongly in my first review (missed by the authors?). As the authors do or should know, the vertebrate Pdx protein is, when assayed by itself, a weak transcriptional activator of luciferase constructs that are driven by vertebrate insulin-gene cis-regulatory regions. But, when cotransfected with a bHLH factor such as NeuroD there is a much more robust response. It could be that this species does not use Pdx:(bhlh)-type cooperativity to activate its ILP4 gene, but I would be surprised. And if the authors did the experiment to test this idea, and it did not show stronger activation, I would actually find the result interesting in and of itself. Other experiments from the Montminy group focused on the cooperativity between Pdx and Pbx factors. I would suggest that the following are simple, and could be inserted to this m/s. The cgPdx alone could be compared to the vertebrate Pdx1, in both cases +/- NeuroD, and the strength of cgPdx as an activator, in the authors' own hands, could be then tested against vertebrate factors in a vertebrate cell line.

We thank the reviewer for the explanation, which is very helpful. To address the concerns on the relatively low level of Pdx activation activity, we have now taken two further approaches in this revision. First, we screened other cell lines to see if the response varied, and settled on COS7 as the activity is higher than the previously used HeLa cells. In these cells, mouse Pdx1 showed up to 8.60x activation when 100 ng of expression plasmid was used (with no cofactor transfected, supplementary figureS10b). Oyster Pdx showed 5.59x to 33.01x activation when 100 to 400 ng expression plasmid was used (again, with no co-factor; Figure 5 in main text). Second, as suggested we tried using a bHLH co-factor – NeuroD. Indeed, we identified a possible E-box near the oyster A1/A3 Pdx-binding site, which is suggestive of a bHLH involvement. We used the oyster orthologue of NeuroD for these experiments. Both mPdx1 and cgPdx gave higher activity when cgNeuroD was cotransfected. The level was significantly higher than the sum of the individual effect of each TF. The result is suggestive of synergistic relationship between cgPdx and bHLH factors, but we have tried to report these findings cautiously due to the confounding factors such as the heterologous cell system.

The authors claim that "we identify an upstream regulatory element of the oyster ILP4 gene directly bound by Pdx protein in oyster hepatopancreas and demonstrate, using a cell culture assay, that oyster Pdx acts as a transcriptional activator through this site." For the sake of correctness, however, they have produced evidence consistent with this idea, but not demonstrated that it DOES act through this site. There is the issue of the ChIP referred to below, but the heterologous cell-culture luc-reporter assays show that this factor CAN act here (for a minimal effect - more below), rather

than it does act here - again, evidence in SUPPORT of the view that cgPdx does similar things to cgILP4 that vertebrate Pdx1 does to Insulin. This type of text needs to be modified accordingly, and checked elsewhere for this strict interpretation.

Thank you for these suggestions, we fully agree with the caution needed. We have corrected the text accordingly.

In a related vein, I find the overall title of the manuscript to be non-descriptive of what has been discovered, and recommend reworking it.

We have revised the title to a more specific one: "Evidence from oyster suggests an ancient role for Pdx in regulating insulin gene expression in animals".

The main text states that Pdx1 binds target genes in liver - but this is not the case, and it's very misleading. The cited article (#39) is a study of reprogramming of hepatic cells by Pdx1 - not akin to normal liver at all. Pdx1 is not expressed in normal liver cells. Indeed, as well as fixing this issue, they might want to refer to the ability of Pdx1 to show repressive as well as activating activity, context-dependent, but also even in the same cell type (vertebrate mature insulin-producing beta cells). When ESC are differentiated into pancreatic beta cells, there is a stage where Pdx1 begins to exert its repressive influence on other (e.g., liver genes - Teo et al., Stem Cell Reports, 2015).

Thank you for the helpful comments. We have re-written the discussion on Pdx and liver/pancreas.

To be explicit again, the authors should discuss in a detailed manner how little conservation there is to the relatively deeply characterized vertebrate insulin locus - how none or a very few of the motifs are present in a tightly spaced manner in the cg ILP4 cis-regulatory region surrounding their B4 site.

This Discussion has now been added. Three high conserved core elements were identified in the ATAC-seq peak: A-box, E-box and HRE, possibly bound by Pdx, NeuroD and HNF4A. There may also be other binding sites upstream cgILP. Details are discussed in the revised manuscript.

Figure 4 is very hard to follow - it needs some cosmetic alterations. I can't tell what's going on with the ATAC-seq landscape (ATAC-Seq R1) and the bars that are supposed to be indicating the called peaks. There are peaks that are in other exons and in other regions that are not underlain by the blue boxes - why is this - please explain in the legend. Is this correct? If so, the text of the legend needs to explain it. In the enlargement, why does the blue bar come to an end halfway through the main peak at the right hand side? Also: Purely minor cosmetic: more space in between the "* additional Pdx site" and the line underneath, bottom left text of the panel? The four exons of the ILP4 gene are indicated in the top bar? and the TSS is at the left end, transcription to the right? Please indicate.

We apologize for not being clearer in the original figure. We have improved our explanation in the figure legend concerning the ATAC-seq landscape and the bars. Details are as follows:

The mapped and processed binned ATAC-seq signal is represented on top of the figure in a histogram-like fashion as visualised in UCSC genome browser in wig format. The “called peaks” are visualised as the ‘bars’ (bed format in UCSC genome browser) that the reviewer refers to. The peak calling process is not a direct representation of the regions with the highest signal. Instead, peak calling takes into account the averaged background over a window as well as the shift induced by the library construction process. Furthermore, only peaks represented in both replicates were retained (see the following figure), which can also explain why some high-signal regions in one replicate do not result in a consensus peak. The wig and bed format data were submitted to GEO.

Other cosmetic works were also made according to the comments.

While I clearly do understand that there are no cg cultured cell lines or transgenic approaches that would address how the cis-reg regions of IL4 are regulated by Pdx, it is, to me, worth including a "rationale" statement for the reader, where the move to the HeLa cell line is justified as a minimal albeit extremely heterologous context for the ability of cgPdx to mount an activation response.

Explanation has been added as suggested.

The CHIP experiment may - and I emphasize "may" - be flawed. The antibodies were, mistakenly in my view, generated against the entire CDS of cgPdx, leading to potential problems with antibodies being generated against the homeodomain itself, which may share substantial similarity/identity with homeodomains of other proteins present in cg. Therefore the experiment in Fig. 5d might not unequivocally show Pdx occupancy at the B4 site. Agreed the test against B3 shows something binding at the B4-containing PCR amplicon region, which is good. But is it Pdx? Something must be shown about the specificity of the antibodies, and whether or not they react with homeoproteins of other types. While discussing this figure, it is also standard to have tested against non-specific IgG for the degree of enrichment of binding at this region - the result with B3 versus B4 seems good, but could the authors simply show us what happened with the IgG-alone control too, please?

According to comments on the CHIP experiment from reviewers #1 and #3, we re-conducted CHIP-qPCR experiment with two antibodies developed by synthesized antigen from the 1-160 aa of cgPdx (see figure below). We report here the fold enrichment assay results on three possible Pdx binding sites (cgILP, cgPCSK1 and cgXBP1) by comparing with mock (IgG).

I suspect this is a matter of style, and the journal itself may not care much at all, but many parts of

the discussion read rather redundantly or repetitively with the text of the introduction or results sections. In some places, rather too "listy" of all that was done, and for my preference this makes the reading the discussion quite wearing.

We have re-checked the throughout article to avoid redundant words. We removed redundant discussion on Pdx and insulin genes between introduction and discussion sections. We removed the discussion related to insulin characterization and phylogenetics. We have deleted the discussion on non-coding region functional studies. Additional discussion has been added as suggested, e.g. Pdx role in the regulation of pancreas and hepatic genes; differences between oyster and vertebrates insulin regulation elements.

The various tissues in the lower-magnification images of Figure 2 need labeling otherwise it's just not useful to look at. This figure needs some major cosmetic improvement to increase the use to the reader. A very few of the reader population will be oyster experts.

Revised as suggested. We added labels on main tissues, and reduced panels.

Minor comments:

Where "residues" are described as affecting the ability of the core TAAT to bind homeodomain factors, this should refer to nucleotides not (amino acid) residues, I believe.

Corrected.

Some parts of the text read awkwardly. The subtitle of the results text "Identification of non-coding regulatory elements by ATAC-seq" reads oddly, because ATAC-seq just reports open chromatin, not non-coding regulatory elements. I suggest reworking this subtitle. The description of amino acids at several key sites having "similar physicochemical properties" just seems odd again - residues are conservatively substituted? r "residues have similar R-group character"?

Thank you for helpful suggestions. We have change the subtitle to "ATAC-seq analysis of oyster pancreas". We rechecked the residual substitution in the three key sites of Pdx, one of which (C18R) should be radical substitution, from cysteine in vertebrates to tyrosine in the assayed invertebrates (lancet and bivalves). Thanks for reminding and sorry for the mistake.

At the text "Curiously, although, we observe a larger number of..." should the second comma be there? Removal makes it easier to read.

Corrected.

At text "We hypothesized that one of these sites to be a candidate for Pdx-binding site in oyster hepatopancreas. Possible binding sites of other insulin-related transcription factors were also identified around 1,806 bp site peak, including FoxO forkhead transcription factor (at -1675 bp: CTGTTTAC) and nuclear receptors (e.g. HNF4A at -1734 bp: GAGGTCA).", the language precision

has drifted a little - please check for correct grammar and readability.

Corrected.

--

Reviewer #3 (Remarks to the Author):

I know this work because I have reviewed the previous version of this manuscript. I can appreciate that the authors considerably improved the manuscript, mainly by producing an oyster PDX1 antibody and performing chromatin immunoprecipitation experiments to confirm the main point of this paper, i.e., the Pdx gene is upstream of insulin in oyster.

The current manuscript is therefore much more solid now than before and the point is clearly made. However, I still think the manuscript can improve in the presentation of the data, in particular at the figure level, and with some additional analysis. In particular:

- Fig. 1: It is not easy to identify the oyster insulin like peptides in the tree. I would suggest to highlight the names of the oyster ILP1, ILP2, ILP3 and ILP4 in bold and/or different colour to spot them at a first glance. Also, please spell out the abbreviation MIP, or at least the M of it, also to be consistent with the other group names (Amphioxus, Nematode, etc)

Corrected as suggested. Oyster ILPs were highlighted with red color. "MIP" was revised to "Molluscs MIP"

- Fig. 2: The current size of the panels is too small; I suggest to arrange the panels in a different way, to have a figure taller than larger, so to have less panels per row and, overall, bigger panels. If there were a size limitation from the journal, then I would reduce the number of panels, transferring some of them in a Supplementary figure

Revised as suggested. We removed three columns images and adjusted the figure panels.

- Fig. 3: same comments as Fig. 2; for this figure, in case, due to space constraints, a reduction of panels were necessary, I would suggest to give a major space to panels (d) and (e) and move (a), (b) and (c) to Supplementary material; in addition, the title of this figure is very little informative; I would use a title which is more indicative of the real content of the figures and the results conceived by it; like it is right now "Assessment of the quality of ATAC-seq data" the figure should be transferred entirely to Supplementary material. Indeed, there is useful info there (at least panel d and e) and these should be highlighted and better visible

Revised as suggested. We edited the title, and retained only original panel d&e. Other panels were moved to supplementary file.

- Fig. 4: also here the title of the figure is very boring, actually this is a necessary description, but

not a title. Please give a title to this figure which says a bit more about its content

Revised as suggested.

- Fig. 5: also here I would change the title, to a less technical description. Do not know, maybe something like “Nuclear localization, transcriptional activity and chromatin immunoprecipitation of oyster Pdx protein” or even more generic, “Nuclear localization and functional characterization of oyster Pdx protein”; also, please better describe what is reported in (d)

Revised as suggested.

- An additional comment here concerns the analysis of the ATAC-seq and ChIP data. The authors state that: “We scanned these accessible chromatin peaks for consensus Pdx motifs obtained through TFBS clustering of five Pdx1 Transcription Factor Binding Sites (TFBSs) derived from the literature (Supplementary Figure S6). A total of 19,104 putative Pdx binding sites were deemed statistically significant within the 168,206 accessible chromatin peaks ($P < 0.05$). The number is comparable with observed binding peaks in human pancreatic islet analysed with ChIPseq (18,294) 28 or ATACseq (~6000) 29.” This is very interesting and is a clear indication that Pdx might have a role in controlling hepatopancreas specific genes in oyster similarly to what the homologous gene does in the vertebrate pancreas. Now, since the authors have produced and analysed a good hepatopancreas transcriptome and since they also have produced oyster Pdx ChIPed material, it would be interesting to know at which level the ATAC seq and RNAseq data overlap; in other words, it would be interesting to analyse how many of the PDX binding site enriched peaks are putatively controlling hepatopancreas DE genes and/or validate a few of these targets by qPCR of the oyster Pdx ChIPed material; this additional analysis, which the authors have probably made already, in my opinion, would add strength to the finding of a conserved transcriptional role of Pdx.

Thanks for the helpful suggestions. We analyzed the ATACseq and RNAseq data of oyster hepatopancreas, and identified 795 oyster hepatopancreas-enriched genes with a putative Pdx binding site bearing ATACseq peaks, of which 56 orthologs have been predicted to be Pdx1 targets in previous high throughput studies on human or mouse (Supplementary TableS6 & Figure S7). We conducted ChIP-qPCR on two of these genes: *cgPCSK1* and *cgXBP1*. The result indicated possible regulation function of *cgPdx* and *cgPCSK1*, further supporting the homology of oyster hepatopancreas to vertebrate pancreas.

Reviewers' Comments:

Reviewer #2:

Remarks to the Author:

Antibody ChIP data are supportive of the conclusions made in the revised version of the m/s. The co-transfection data with the bHLH protein make sense, and yield more substantial evidence of the ability for synergistic transactivation on a reporter construct, thus standing up as similar in quality to accepted already published data in mammals.

The remaining issue I have is with the description of the quality and specificity of the two new cgPdx peptide antibodies that have been raised. It was very dedicated of the authors to run off and make these new reagents. But the detail around their specificity is sparse, not only in the main text, but in the supplemental data/methods that are shown. E9112 yields a band at the approximate expected size, but E9113 mostly reacts with a ~90 kDa species, with very little indeed at ~50 kDa. It would also be helpful in the methods to simply list the a.a. sequence for the two peptides, and not have to uncover them from the whole protein sequence. The relevance of the PARP1 Ab blot is not given. Figure 5 is scrunched together too much, leaving the labels (e.g., cgNeuroD quantities (ng)) hard to associate with the correct panel. Also no specific notes are given for d, e and f in this figure. The HP samples 1 and 2 need to stand out clearer within these panels. Yet, related to the western blot data on 9113 Ab having the poorest selectivity, it gives by far (in notable cases) the best ChIP enrichment over mock IgG. I suppose it's sufficient as it stands, but I might suggest that the ChIP'd fragments for 9113 could be sequence-verified as to coming exactly from the expected region in this PCR. [If that was done, I could not find the details.] Another standard procedure is to run EMSA as part of the validation of the antibody specificity. This is such an important part of the paper, and the data look very promising, that this is the only part with which I have a lingering issue. If the authors decide to do no more here, the caveats must be described and dealt with in the text much better.

Typos noted: the insulin gene, withits activity is, Digoxigenin-labeled, was used to validation antibodies, molecular weight for cgPdx was produce,

Reviewer #2 (Remarks to the Author):

Antibody CHIP data are supportive of the conclusions made in the revised version of the m/s. The co-transfection data with the bHLH protein make sense, and yield more substantial evidence of the ability for synergistic transactivation on a reporter construct, thus standing up as similar in quality to accepted already published data in mammals.

The remaining issue I have is with the description of the quality and specificity of the two new cgPdx peptide antibodies that have been raised. It was very dedicated of the authors to run off and make these new reagents. But the detail around their specificity is sparse, not only in the main text, but in the supplemental data/methods that are shown. E9112 yields a band at the approximate expected size, but E9113 mostly reacts with a ~90 kDa species, with very little indeed at ~50 kDa.

Response: We thank the reviewer for the positive comments. We have added more text in the manuscript concerning validation and specificity of the two antibodies as suggested:

1. **In the method section:** Validation of antibodies was conducted by western blotting on the tissue lysate of oyster hepatopancreas (Supplementary Fig. 11). The full coding sequence of cgPdx was cloned in-frame (see Supplementary Data 4 for primers) with a V5 tag at the C-terminal (pSF-CMV-Puro-COOH-V5, Oxford Genetics) and expressed in HEK293T cells. Western blotting was then performed with the cell lysate to determine the molecular weight of cgPdx. Specificity of antibodies was tested by western blotting on oyster hepatopancreas lysate; anti-PARP1 antibody (ab32138, abcam, Shanghai, China) was used as a positive control.
2. **In the discussion section:** However, the evidence for protein-binding *in vivo* is best treated as indicative rather than conclusive until the newly generated antibodies used in this study are further validated⁵⁵. For example, although western blots using E9112 and E9113 showed a signal around the expected cgPdx molecular weight, other bands are also detected, especially with E9113.

It would also be helpful in the methods to simply list the a.a. sequence for the two peptides, and not have to uncover them from the whole protein sequence. The relevance of the PARP1 Ab blot is not given.

Response: We used the full-length peptide of 1-160 a.a. as antigen to produce antibodies E9112 and E9113 by immunizing two rabbits. The sequence is now shown in Supplementary Fig. 3. Raw images for the western blot (including PARP1 Ab blot) are provided in Source Data file.

Figure 5 is scrunched together too much, leaving the labels (e.g., cgNeuroD quantities (ng)) hard to associate with the correct panel. Also no specific notes are given for d, e and f in this figure. The HP samples 1 and 2 need to stand out clearer within these panels.

Response: We have increased the space between panels to make labels clearer. Notes on d, e and f were added, and hepatopancreas samples Hep1 and Hep2 were marked clearly on them.

Yet, related to the western blot data on 9113 Ab having the poorest selectivity, it gives by far (in notable cases) the best CHIP enrichment over mock IgG. I suppose it's sufficient as it stands, but I might suggest that the CHIP'd fragments for 9113 could be sequence-verified as to coming exactly from the expected region in this PCR. [If that was done, I could not find the details.] Another standard procedure is to run EMSA as part of the validation of the antibody specificity. This is such

an important part of the paper, and the data look very promising, that this is the only part with which I have a lingering issue. If the authors decide to do no more here, the caveats must be described and dealt with in the text much better.

Response: We have not been able to perform sequencing verification of ChIP'd fragments. We agree that this caveat needs to be clear and have added a paragraph in the Discussion section to specifically note this. Specifically, we have added the phrase "However, the evidence for protein-binding *in vivo* should be treated as indicative rather than conclusive until the newly generated antibodies used in this study are further validated" – as noted above.

Typos noted: the insulin gene, withits activity is, Digoxigenin-labled, was used to validation antibodies, molecular weight for cgPdx was produce,

Response: Thank you, all typos have been corrected (plus a few others we found in the manuscript).